# Stimulation of tumoricidal immunity via bacteriotherapy inhibits glioblastoma relapse

Yulin Zhang[1,2,3,10], Kaiyan Xi[1,2,4,10], Zhipeng Fu[3,10], Yuying Zhang[5], Bo Cheng[6], Fan Feng[1,2], Yuanmin Dong[3], Zezheng Fang[1,2], Yi Zhang[1,2], Jianyu Shen[1,2], Mingrui Wang[1,2], Xu Han[1,2], Huimin Geng[1,2], Lei Sun[7], Xingang Li ![ORCID][1,8,9], Chen Chen[3,11] ✉, Xinyi Jiang ![ORCID][3,11] ✉ & Shilei Ni ![ORCID][1,2,11] ✉

Glioblastoma multiforme (GBM) is a highly aggressive brain tumor characterized by invasive behavior and a compromised immune response, presenting treatment challenges. Surgical debulking of GBM fails to address its highly infiltrative nature, leaving neoplastic satellites in an environment characterized by impaired immune surveillance, ultimately paving the way for tumor recurrence. Tracking and eradicating residual GBM cells by boosting antitumor immunity is critical for preventing postoperative relapse, but effective immunotherapeutic strategies remain elusive. Here, we report a cavity-injectable bacterium-hydrogel superstructure that targets GBM satellites around the cavity, triggers GBM pyroptosis, and initiates innate and adaptive immune responses, which prevent postoperative GBM relapse in male mice. The immunostimulatory Salmonella delivery vehicles (SDVs) engineered from attenuated *Salmonella typhimurium* (VNP20009) seek and attack GBM cells. Salmonella lysis-inducing nanocapsules (SLINs), designed to trigger autolysis, are tethered to the SDVs, eliciting antitumor immune response through the intracellular release of bacterial components. Furthermore, SDVs and SLINs administration via intracavitary injection of the ATP-responsive hydrogel can recruit phagocytes and promote antigen presentation, initiating an adaptive immune response. Therefore, our work offers a local bacteriotherapy for stimulating anti-GBM immunity, with potential applicability for patients facing malignancies at a high risk of recurrence.

Glioblastoma multiforme (GBM) is the most aggressive and common malignancy and has a grim prognosis in patients undergoing surgery followed by radio- and chemotherapy[1–3]. The highly invasive nature of GBM and the associated impaired immune response leads to the intermingling of tumor cells with the normal brain parenchyma, making complete resection of GBM impossible[4–6]. Tracking and eliminating residual GBM cells after surgery by boosting antitumor immunity is critical for inhibiting the recurrence of this fatal disease, but effective interventions remain elusive.

Macrophages and microglia are the most abundant innate immune cells in the GBM microenvironment, accounting for 30% to 50% of all components[7,8]. Surgical resection recruits phagocytes to the margin of the cavity, accompanied by the release of inflammatory factors, cytokines, and growth factors[9,10]. The reprogramming

of macrophages creates a favorable immune microenvironment for postoperative recurrence and causes relapse in more than 90% of patients after surgery[11–13]. In light of the unique function and abundance of phagocytes, we hypothesized that activation of the innate immune function of phagocytes could counteract GBM cells and enable antigen presentation to activate adaptive immunity and prevent postoperative relapse.

Microorganisms, especially gram-negative bacteria, have the capacity to trigger antitumor immune responses by engaging Toll-like receptors (TLRs) and nucleotide-binding oligomeric receptors (NLRs) of phagocytes[14–17]. Notably, the stimulation of innate immunity by bacteriotherapy has emerged as an effective strategy for mitigating the tumor-specific immunosuppressive microenvironment[18,19]. Intracranial bacterial infections promote immune cell recruitment and activation, and bacteria-based immune cell recruitment strategies are feasible for overcoming the "cold" microenvironment of GBM[20,21]. GBM-infiltrating bacteria can also activate adaptive immunity by presenting tumor-associated antigens to T cells via bacterial peptides[22]. Furthermore, retrospective studies have shown a correlation between localized gram-negative bacterial infections after GBM surgery and improved patient prognosis[23–25]. On this basis, we proposed a bacteriotherapy utilizing bacteria to activate local innate and adaptive immunity in the surgical cavity to curb postoperative relapse of GBM.

Attenuated *Salmonella typhimurium* (VNP20009) is unique due to its dual ability to activate both innate and adaptive immunity[26,27]. However, investigations of intracranial bacteriotherapy have been limited, primarily due to concerns about inducing infection by employing bacteria[28,29].

Here, we show an immunostimulatory autolysing Salmonella-nanocapsule delivery system (IASNDS) by tethering Salmonella lysis-inducing nanocapsules (SLINs) to the surface of a Salmonella delivery vehicle (SDV). An ATP-responsive hydrogel is employed to deliver the IASND into the surgical cavity and synergistically enhance immune activation. The hydrogel-delivered IASNDS can activate the innate immune response and remodel the GBM immune microenvironment by liberating bacterial components within GBM cells, coupled with enhanced phagocytosis and antigen presentation, which activate the adaptive immune response. These results provide a promising strategy to prevent postoperative recurrence of GBM and warrant further evaluation in clinical trials.

## Results

### Design of Salmonella delivery vehicles targeted to the GBM microenvironment

Three essential components are required for intracranial administration of bacterial delivery systems: (1) accurate targeting of the tumor microenvironment, (2) effective cell invasion, and (3) excellent safety and controllability. Safety is a prerequisite for the successful application of autolyzing SDVs in the treatment of intracranial disorders. The Salmonella strain VNP20009 (with $\Delta msbB$, $\Delta purI$, and $\Delta xyl$ deletions) is a well-known bacterial delivery system, but it is not safe enough for intracranial delivery due to the distinctive features of intracranial immunity[30,31]. We designed two genetic circuits that can activate invasion protein A (invA) expression under hypoxic conditions to enhance SDV invasion and induce Lysin E (LysE) expression, which causes bacterial lysis and subsequent release of lysogenic components in GBM cells (Fig. 1a). Exploiting the potential of P-selectin on the surface of megakaryocyte-derived exosomes to bind to CD44, which is highly expressed on the surface of GBM cells, we achieved precise active targeting to tumor cells (Supplementary Fig. 1). In pursuit of a synergistic therapy employing SDVs, we orchestrated the expression of exosome-localized brain acid soluble protein 1 (BASP1) and gasdermin D (GSDMD), where BASP1 anchors the GSDMD to the surface of the exosome membrane. Further, the construction of SLINs achieved by encapsulating L-arabinose in GSDMD-EXO through electroporation,

SLINs can be degraded intracellularly releasing L-arabinose which could further induce SDVs to express LysE. This fusion protein was strategically positioned on the exosomal membrane, followed by coupling of the exosomes with the SDVs surface via an azide-alkyne cycloaddition reaction. This strategy culminated in the creation of an IASNDS, as depicted in Fig. 1b. We next drew inspiration from the model of phage-mediated bacterial lysis to strengthen this conceptual framework. L-arabinose is added to exosomes to form SLINs, which initiate SDV autolysis. The structural configuration not only guarantees the safety of intracranial SDVs application but also augments the capacity of SDVs to induce immunogenic cell death (ICD). This, in turn, increases the infiltration of immune cells within the GBM microenvironment and amplifies the innate immune response associated with the SDV system.

The rapid proliferation of GBM cells occurs simultaneously with the delayed development of the tumor vasculature, creating a hypoxic environment within the tumor[32,33]. The hypoxic state frequently precipitates necrotic tissue formation in the central region of the GBM tumor, thereby establishing a distinctive hallmark that sets it apart from other benign intracranial tumors. In this study, to investigate the capability of SDVs to invade GBM cells, we discovered that autolysing SDVs labeled with a green fluorescent protein (GFP+) proliferated and generated Salmonella-containing vesicles (SCV) in GBM cells (Supplementary Fig. 2). To further validate the response of the SDVs to hypoxia (1% oxygen), the efficacy of the SDVs (GFP+) was subsequently investigated in hypoxic and normoxic (21% oxygen) environments, and the results showed that under hypoxic conditions, the survival rate of the SDVs was increased and the fluorescence intensity was enhanced (Fig. 1c, d).

To gain insight into the spatial distribution of SDVs in a syngeneic orthotopic mouse GBM model, we intravenously injected SDVs and VNP20009 (Fig. 1e, f). Compared to VNP20009, the SDVs (red) showed significant aggregation at the tumor site, and dispersion within normal brain tissue was not evident (white arrows, Fig. 1g, bottom). The mechanism of SDVs accumulation and survival in GBM is due to the introduction of *FNR-invA*, and the local immunosuppressive microenvironment also inhibited the immunological clearance of SDVs. The observed increased porosity in tumor tissue within the SDVs group seems to be associated with the occurrence of pyroptosis in GBM cells, leading to clearance by the immune system. In contrast to the parental Salmonella strain devoid of Lysin E (LysE⁻), the SDV engineered to express Lysin E (LysE⁺) from bacteriophage phiX174 induced lysis and promoted GFP release into the tumor cytoplasm upon exposure to L-arabinose (Fig. 1h). Notably, the half maximal effective concentration (EC50) of L-arabinose was determined to be 91.5767 μM (Supplementary Fig. 3). In summary, these findings imply that our designed SDV is capable of targeting the hypoxic region of the GBM and inducing lysis to release bacterial components.

### Design of Salmonella lysis-inducing nanocapsules

Pyroptosis, an important form of ICD, plays a vital role in the fight against infections and endogenous danger signals[34,35]. Cytotoxic lymphocytes rely on gasdermin-mediated pyroptosis to kill tumor cells, suggesting that pyroptosis is also intimately involved in anticancer immune responses. Among the gasdermin family proteins, the membrane perforin gasdermin D (GSDMD) serves as a potent executor of pyroptosis, and its functions and mechanisms of action in the activation of immune cells have been demonstrated[36,37]. Here, GSDMD was attached to the exosome surface by the exosome membrane protein BASP1 to increase the content of GSDMD in tumor cells (Fig. 2a). A significant increase in GSDMD levels in megakaryocytes (MEG01) (an approximately 9.94-fold increase) was observed by PCR analysis (Fig. 2b). This increase in GSDMD protein expression was substantiated further through Western blotting (Fig. 2c). Subsequent harvesting of exosomes produced by MEG01 corroborated the presence of

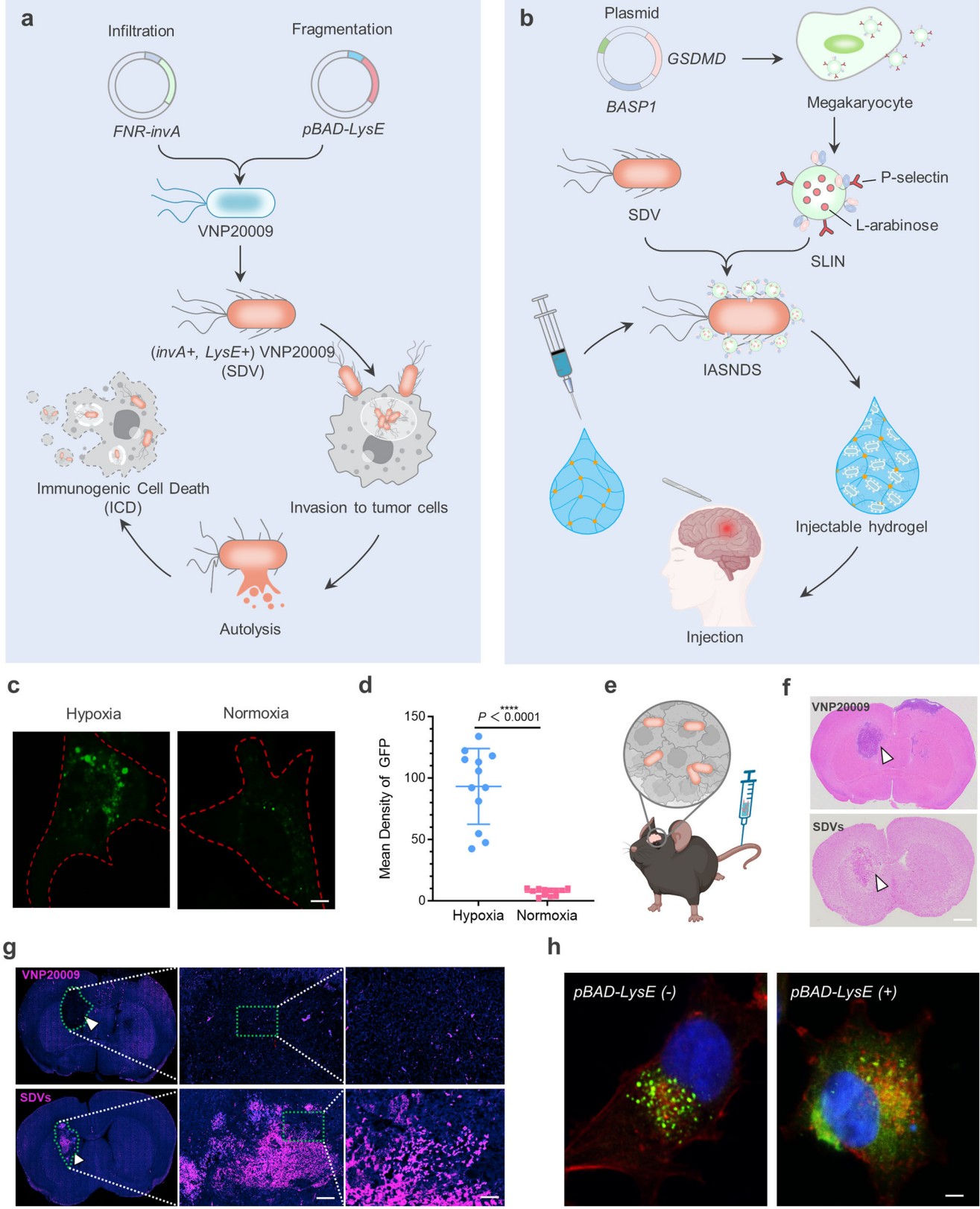

exosome-specific proteins on the surface of GSDMD-EXOs and demonstrated overexpression of GSDMD (Fig. 2d, e). Comparing the difference between SDV equipped with GSDMD (+) SLIN and GSDMD (−) SLIN, respectively, we found that GSDMD (+) SLIN in combination with SDV could significantly increase cellular pyroptosis (Supplementary Fig. 4).

The pivotal step for the release of immune-activating components from SDVs depends on the initiation of autonomous lysis after cellular invasion. The release of L-arabinose in GBM cells induced the activation of the pBAD promoter within the SDV, which initiated the LysE expression. This circuit was achieved by encapsulating L-arabinose in GSDMD-EXO via electroporation, forming nanocapsules named SLINs

**Fig. 1 | Mechanism, preparation, and characterization of the hydrogel-based autolysing bacterial delivery system. a** Schematic depicting the strategic transformation of Salmonella into the SDV and the induction of ICD in GBM cells via Salmonella VNP20009. **b** Illustration detailing the preparation of the IASNDS and synergistic localized treatment with the hydrogel for surgical cavity application. **c** Visualization of fluorescence (GFP⁺) in the SDV (green) under both normoxic and hypoxic conditions for 24 hours. Red dashed lines represent GL261 cell boundaries. (n = 3 independent experiments). **d** Statistical analysis showing the intensity of GFP fluorescence in normoxic and hypoxic cultures. (n = 12 images from three independent experiments). (exact P value: P = 2.42423E-09); ****P < 0.0001. **e, f** In vivo distribution assessment of 1 × 10^7 CFU SDVs (GFP) injected into intracranial tumor-bearing mice via the tail vein. Brain GBM tissue sectioning was performed, with the SDVs depicted by a white arrow. **g** Additional fluorescence staining of brain tissue sections to observe the SDV distribution (GFP⁻, shown in red) in intracranial GBM tissues (white arrows). (n = 3 independent experiments). **h** Visualization of the green fluorescence distribution of the SDVs (GFP⁺) under *pBAD-LysE* (-) and *pBAD-LysE* (+) conditions induced by 500 µM L-arabinose for 24 hours in the cell culture medium (n = 3 independent experiments). The intracellular GFP distribution in the *pBAD-LysE* (+) condition indicates initiation of SDV autolysis in GL261 cells. Data are shown as the means ± SEMs. The statistical comparisons in **d** were performed with two-tailed, unpaired Student's *t* tests, with asterisks indicating significant differences. **c, f, g,** and **h** show representative images of the corresponding independent biological samples. (n = 3 independent biological samples). The scale bar in **c** is 1 µm, the scale bar in **f** is 1.5 mm, the scale bars in **g** are 50 µm (middle) and 10 µm (right), and the scale bar in **h** is 1 µm. Source data are provided as a Source Data file.

with 4.82% drug loading capacity measured by HPLC. Further characterization of these SLINs revealed that their morphology did not change significantly after loading with L-arabinose, and the particle size was approximately 100 nm (Fig. 2f, g). SLINs were able to maintain relatively good stability over one week, and the content of L-arabinose remained in the desirable range (Fig. 2h). SLINs were placed in dialysis bags with a diameter of 10 nm and incubated with 1 × 10^7 bacterial CFU (Colony-Forming Units) to observe their stability, and their size did not change significantly within one week (Fig. 2i). Hence, SLINs are highly stable prior to their invasion of GBM cells, without the early release of the loaded L-arabinose.

## The IASNDS invades GBM cells and initiates autolysis
Free GSDMD does not diffuse into GBM cells mainly due to its large molecular weight and negative surface charge, and GSDMD is unable to trigger cellular pyroptosis owing to concealment of its pore-forming domains[38]. To address this challenge, *Salmonella typhimurium* was used for intracellular delivery of GSDMD. Due to the properties of the intracellular bacterium in the SDV in the IASNDS, the SLINs on its surface are brought inside the GBM cell, and after inducing pyroptosis, they further activate innate and adaptive immunity to combat postoperative relapse (Fig. 3a).

The IASNDS arrives in the cell and forms Salmonella-containing vesicles (SCVs), which can rapidly multiply and prevent cellular clearance (Supplementary Fig. 5). SLINs can be degraded by GBM cells and release GSDMD and L-arabinose, slowly initiating the SDV lysis process, which can further activate caspase 1 to cleaved caspase 1. Then, cleaved caspase 1 converts GSDMD into pore-forming domain N-terminal GSDMD to bind to the GBM membrane and subsequently trigger cellular pyroptosis (Fig. 3b). As a type of ICD, pyroptosis releases tumor antigens, cytokines, and chemokines that significantly activate antitumor immune responses.

Click chemistry was used to achieve efficient coupling of SLINs to the SDV surface (Fig. 3c). The SDV surface was carefully functionalized with azido-PEG4-NHS ester, facilitating subsequent coincubation with DBCO-C6-NHS-modified SLINs. An average of approximately 97 SLINs are bound to the surface of each SDV by this coupling method. Tethering SLINs to the surface of the SDV can catalyze the cleavage process of the SDV by enabling the intake of large amounts of L-arabinose when the IASNDS enters GBM cells (Fig. 3d, e). Dynamic monitoring of the green fluorescence intensity in the cells revealed that when cocultured with 1 ×10^8 CFU of the IASNDS, the SDV slowly initiated the autocleavage process in the cells within 24 hours, with increasing fluorescence intensity of GFP (Fig. 3f, g, Supplementary Fig. 6). Bafilomycin A1 was employed to inhibit lysosomal function, the result showed that lysis of SDVs was significantly inhibited, thus suggesting that degradation of SLINs is achieved via the lysosomal pathway (Supplementary Fig. 7). After approximately 48 h of coculture, the intensity of green fluorescence in the cells reached a maximum, which indicated complete autolysis of the SDV; at the same time, there was a large amount of cytoplasmic leakage in the GBM cells, confirming the formation of pores on the cell surface after the polymerization of GSDMD-N (Fig. 3h, Supplementary Fig. 8).

## GBM cell pyroptosis and antitumor immune activation
Pyroptosis of GBM cells is a vital step in the recruitment and activation of antitumor immune cells and is central to the prevention of GBM recurrence after surgery. We found that the IASNDS was indeed able to induce pyroptosis in GBM cells, and the GBM cells continued to expand and form balloon bubbles until the cell membrane ruptured (Fig. 4a, b, Supplementary Fig. 9). 3D tumor spheres and neurospheres were designed to reveal changes in the tumor invasion behavior of the IASNDS (Fig. 4c). Relative to that of both the control and SLIN-treated groups, the invasiveness of the GBM spheres was markedly attenuated following 24 hours of IASNDS treatment. This manifested as a 50.8% and 61.4% reduction in invasion distance, respectively, compared to that of the control group (Fig. 4d, e, Supplementary Fig. 10).

SYTOX Green was used for staining to evaluate the integrity of the cell membrane. When cells undergo pyroptosis, the permeability of the cell membrane increases and SYTOX binds to nucleic acids and exhibits a significant increase in fluorescence intensity. As shown in Fig. 4f, g, the SYTOX fluorescence intensity was significantly increased after administration with the IASNDS treatment, with 15.14, 2.25, and 2.71 times higher than those of the Ctrl, SDV, and SLIN groups, respectively. In addition, propidium iodide (PI) staining further confirmed that the cells underwent programmed death and exhibited cell membrane damage (Fig. 4h, Supplementary Fig. 11).

The therapeutic mechanism of action of the IASNDS involves the activation of GSDMD, which catalyzes the formation of pores on the cell surface and results in cellular swelling, ultimately facilitating the release of cellular contents. To validate this, our assessment included the quantification of lactate dehydrogenase (LDH) levels within the culture medium. IASNDS treatment caused an approximately 3.54-fold increase in LDH levels relative to those of the SDV group (Supplementary Fig. 12). Notably, elevated secretion of high mobility group box 1 (HMGB1) and adenosine triphosphate (ATP) after IASNDS treatment signified the occurrence of ICD within tumor cells (Fig. 4i, j). Further studies showed that IASNDS treatment caused a significant increase in the levels of cytokines and chemokines associated with immune activation of THP-1 cells after co-incubation with QL01#GBM cells (Fig. 4k). In conclusion, the pronounced amplification of the signaling cascade induced by this bacteriotherapy underscores the efficacy of the IASNDS system in recruiting and activating a variety of immune cells, including phagocytes, natural killer cells, and T cells.

## Smart hydrogels enable the release of ATP-responsive CpG oligonucleotides
Since the brain tissue is filled with flowing cerebrospinal fluid which is unfavorable to the local application of the drug, hydrogel as a carrier for local drug delivery can ensure the sustained release in the operative

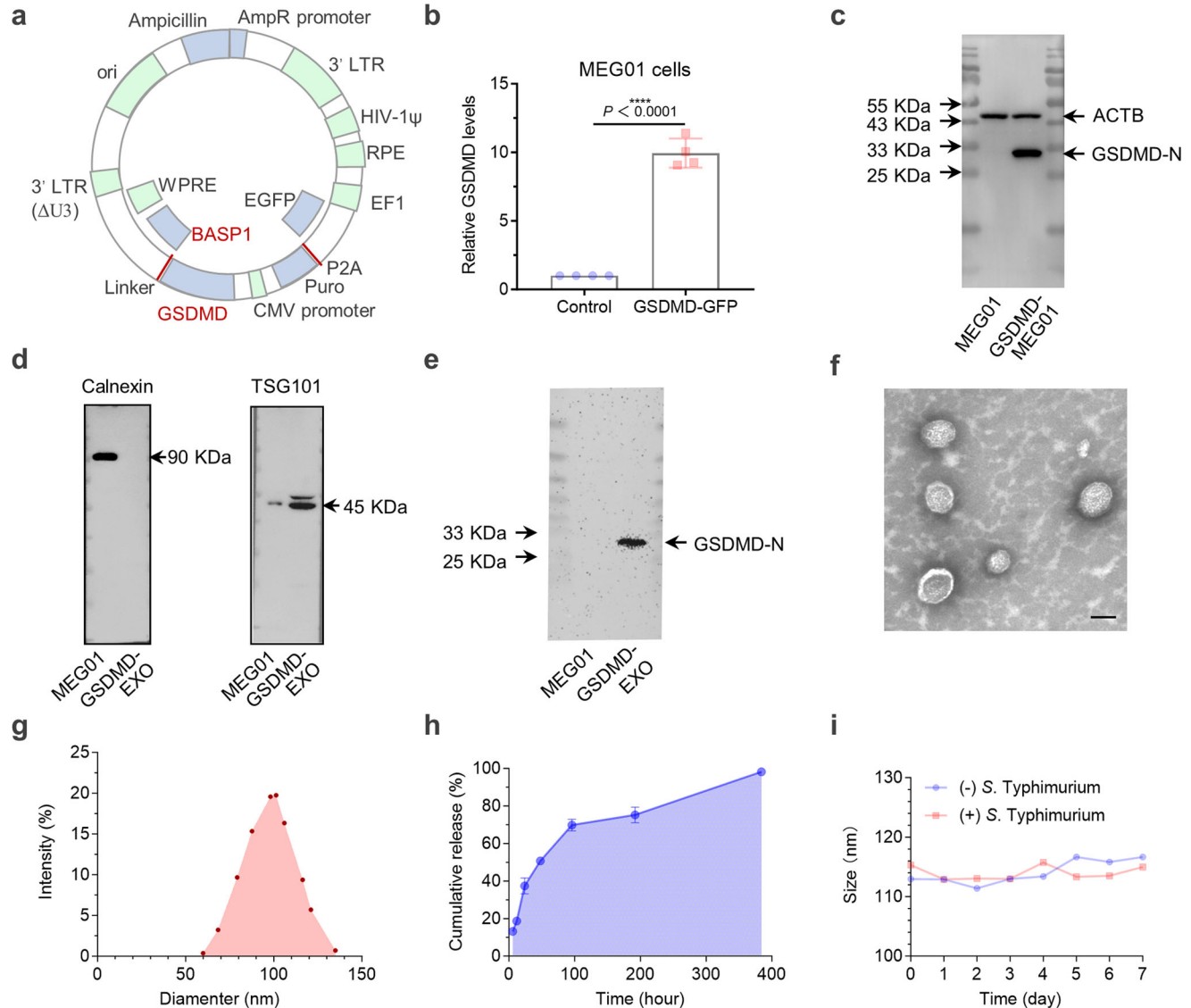

**Fig. 2 | Design and characterization of Salmonella lysis-inducing nanocapsules.** **a** Design of the gene construct illustrating GSDMD overexpression and attachment to the exosome membrane surface. **b** Verification of GSDMD overexpression to evaluate transfection efficiency in the MEG01 cell line through PCR ($n = 4$ independent experiments). (exact $P$ value: $P = 2.75356E\text{-}06$); ****$P < 0.0001$. **c** Western blot analysis showing GSDMD-N protein expression in MEG01 (GSDMD⁺) cells ($n = 3$ independent experiments). **d** The collected exosomes were confirmed to express exosome-specific proteins by Western blotting ($n = 3$ independent experiments). **e** Western blot results confirming GSDMD-N protein expression in exosome-derived (GSDMD-EXO) samples ($n = 3$ independent experiments). **f** Transmission electron microscopy reveals the SLIN morphology and particle size distribution (**g**) using dynamic light scattering (DLS). ($n = 3$ independent experiments). **h** Kinetic profile of L-arabinose release from SLINs in PBS, pH 7.4 ($n = 3$ independent experiments). **i** Particle size changes of SLINs loaded in a 10 nm dialysis bag and coincubated with $1 \times 10^6$ CFU SDVs for 7 days at 4 °C. ($n = 3$ independent experiments). Data are presented as the mean ± S.D. The statistical comparisons in **b** were performed with two-tailed, unpaired Student's $t$ tests, with asterisks indicating significant differences. **c**–**f** show representative images of the corresponding independent biological samples. The scale bar in **f** is 50 nm. Source data are provided as a Source Data file.

cavity, avoiding the toxicity to the normal brain tissue. Intracellular SDVs curtail the extracellular release of nucleic acids after bacterial lysis, potentially posing challenges to achieving enduring immunotherapeutic effects with the SDVs. This limitation implies that the activation of adaptive immunity may not be durably sustained once host cells are cleared by phagocytes. In light of these considerations, we devised a strategy involving an immune-activating hydrogel (termed "Gel") to couple CpG ODN with a nucleic acid aptamer. This approach capitalizes on the release of ATP resulting from IASNDS-induced cellular pyroptosis, thereby fostering sustained activation of immune cells (Fig. 5a). Notably, the gel, which is cross-linked using UV irradiation, confines the gel to the surgical cavity, effectively

circumventing potential interference with cerebrospinal fluid reflux and mitigating potential side effects.

The initial phase of this endeavor involved the integration of an ATP-specific aptamer (Apt) into the molecular structure of hyaluronic acid methacryloyl (HAMA) via an amide bond. The linkage of Apt and HAMA was evaluated via polyacrylamide gel electrophoresis (PAGE), which demonstrated a significant band shift when the molar ratio of Apt to HAMA reached 12:1 (Fig. 5b). Subsequently, the binding efficacy of HAMA-Apt and CpG was assessed, revealing heightened efficacy at a 1:8 ratio of the two components (Fig. 5c). Further evaluation of the properties of the IASNDS-loaded gel revealed its capacity to maintain remarkable porosity and robust gel formation (Fig. 5d, e). After

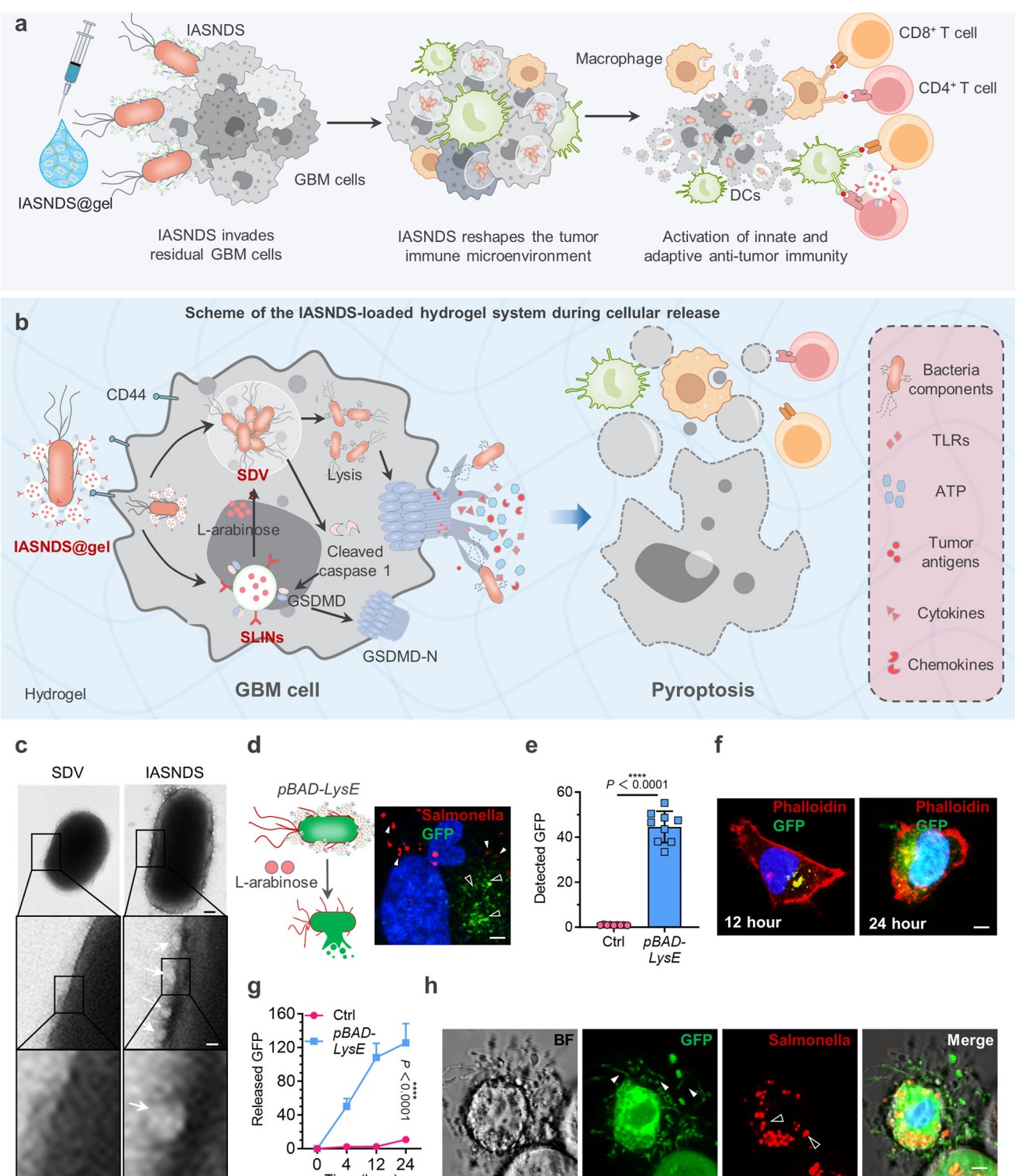

10 seconds of UV irradiation, both the G′ (storage modulus) and G″ (loss modulus) values exhibited notable time-dependent increases. Notably, the G′ value surpassed the G″ value by a factor of approximately ten, signifying successful gel formation (Fig. 5f, g).

The double-stranded structure of the gel can dissociate in the presence of ATP due to Black Hole Quencher 3 (BHQ3, a quencher of Cy5.5) and the proximity of CpG-Cy5.5 to the quenching reaction, suggesting that the CpG ODN can be released in response to elevated ATP levels (Fig. 5h). Further studies confirmed that CpG ODN could

indeed be liberated in the presence of ATP, and the responsiveness of gel release was positively correlated with the ATP concentration (Supplementary Fig. 13). Notably, the addition of various ATP concentrations to the solution post gel formation yielded distinct fluorescence intensity variations, and higher concentrations of ATP accelerated the release of CpG ODN (Fig. 5i, j). Co-incubation of ATP-responsive hydrogel in artificial cerebrospinal fluid simulating the in vivo environment revealed that the degradation was related to the ATP concentration, with the higher the concentration the faster rate of

**Fig. 3 | Mechanism of immunogenic cell death induced by the autolysing SDV system. a** Illustration depicting the IASNDS system invading GBM cells, triggering cell death, activating antigen-presenting cells, and eliciting an immune response. **b** IASNDS cellular entry, intracellular autocleavage initiation, and tumor cell pyroptosis induction mechanism. **c** Transmission electron microscopy characterizing the structures of the SDVs and IASNDS, highlighting SLINs coupled to the IASNDS surface (white arrow). (*n* = 3 independent experiments). **d** GFP release from SDVs into the cytoplasm (green, black arrow) and the presence of lysed Salmonella membranes inside cells (red, white arrow) after coculture of 1 × 10^6 CFU of the IASNDS with QL01#GBM cells for 48 hours (*n* = 3 independent experiments). **e** Detection of the GFP fluorescence intensity inside tumor cells after coincubation of 1 × 10^6 CFU of the IASNDS with QL01#GBM cells for 48 hours (*n* = 9 images from three independent experiments). (exact *P* value: *P* = 2.491E−12); ****P* < 0.0001.

**f, g** Intratumor cell fluorescence intensity over 24 hours in QL01#GBM cells, illustrating IASND intracellular release behavior (*n* = 9 images from three independent experiments). (exact *P* value: *P* = 7.8029E-11); ****P* < 0.0001. **h** Altered cell membrane permeability and cytoplasmic efflux post IASNDS treatment of QL01#GBM cells (green, white arrow), along with dispersed bacterial membranes after SDV cleavage (red, black arrow). (*n* = 3 independent experiments). Data are presented as the mean ± S.D. The statistical comparisons in **e** was performed with two-tailed, unpaired Student's *t* tests, with asterisks indicating significant differences. **c, d, f,** and **h** are representative images of the corresponding independent biological samples. The scale bars in **c** are 150 nm (middle) and 50 nm (bottom), the scale bar in **d** is 0.5 μm, and the scale bars in **f** and **h** are 1 μm. Source data are provided as a Source Data file.

degradation (Supplementary Fig. 14). Furthermore, investigations involving coincubation of gels and macrophages revealed that solutions containing CpG ODN induced the upregulation of TLR4, TLR5, and TLR9 (Supplementary Fig. 15). In conclusion, our synthesized gel effectively promotes the progressive release of CpG ODN in response to the corresponding ATP release triggered by pyroptosis, and the gel can activate the innate immune response.

## Cavitary injection of IASNDS@gel to prevent tumor recurrence

The efficacy of the antitumor intervention in vivo was evaluated utilizing the GL261 syngeneic orthotopic mouse model. Ten days after inoculation of a Luci+GL261 cell suspension, tumor-bearing mice were subjected to random grouping and received various treatments after surgical resection (Fig. 6a). To establish the GBM excision mouse model, a portion of the intracranial GBM in mice was removed with forceps under a microscope on day 10, and then the tumor was suctioned by using a suction device to form a cavity that could accommodate 30 μL volume, hydrogel with different formulations were subsequently injected into the cavity (Supplementary Figs. 16 and 17). During surgical resection, distinct formulations, including Ctrl (PBS), Gel, SLIN@gel, SDV@gel, and IASNDS@gel, were administered to the respective groups (Fig. 6b).

Evaluation of the brain by hematoxylin and eosin (H&E) staining and immunofluorescence after surgery revealed an infiltration of GBM cells into the postsurgical parenchyma and a distribution of these cells around the margins of the cavity, with limited intracranial dissemination (Fig. 6c, d). Dynamic monitoring of tumor volume via bioluminescence imaging (BLI) of Luci+GL261 cells with various treatment intervals underscored the significant inhibition of tumor volume growth upon IASNDS@gel treatment (Fig. 6e, f). Without Gel, SDV or IASNDS alone did not demonstrate significant GBM suppression, primarily because drugs injected locally tend to be rapidly cleared with cerebrospinal fluid circulation. These findings imply that there is a substantial preventative effect of intracavitary injection of IASNDS@gel on GBM recurrence.

Comparative analysis revealed that the gel-alone treatment notably extended the survival of mice (by 2.14-fold) relative to the Ctrl group (Fig. 6g). However, the difference in survival time between the Gel and SLIN@gel groups was not statistically significant. In contrast, the survival time of mice receiving SDV@gel treatment was a remarkable 4.46 times longer than that of the Ctrl group. After up to 120 days of follow-up, 80% of the mice remained alive in the IASNDS@gel group. Immunohistochemical staining of tumor tissues after 30 days of treatment showed Ki-67 expression upon gel-based intervention, with the lowest tissue expression observed in the SDV@gel and IASNDS@gel groups (Fig. 6h, Supplementary Fig. 18). Further fluorescence staining of tumor tissues revealed a narrower distribution and increased vacuole formation in the IASNDS@gel group. These changes and the observed softening of the tumor texture seem to correlate with pyroptosis-mediated tumor cell death (Fig. 6i). These alterations may facilitate the infiltration of immune cells,

consequently reshaping the immune microenvironment of GBM and potentially bolstering the immunotherapeutic effect.

## Mechanisms of anti-GBM effects in vivo

Next, we explore the anti-GBM mechanisms of different therapeutic agents. GBM-bearing mice were euthanized, and tumor tissues were isolated on the 20th day (Fig. 7a). Notably, the excised tumor tissues from mice in the gel treatment group exhibited a marked increase in the number of infiltrated immune cells compared to that in the Ctrl group, suggesting a role of the gel in immune cell recruitment (Fig. 7b, Supplementary Fig. 19). In the IASNDS@gel group, the abundance of CD4+ T cells was significantly increased, with values 6.62, 4.05, 2.95 and 1.87 times higher than those in the Ctrl (PBS), Gel, SLIN@gel, and SDV@gel groups, respectively (Fig. 7c). Encouragingly, CD8+ T cells also showed the same trend, with a notable increase in the number of CD8+Granzyme B+ T cells (Fig. 7d, e, Supplementary Fig. 20). Remarkably, the IASNDS@gel group exhibited a substantial proportion (71.6%) of Granzyme B+ T cells among CD8+ T cells. These findings suggest that IASNDS@gel plays a pivotal role in initiating antitumor immune responses.

As mediators of innate and adaptive immunity, antigen-presenting cells (APCs) play a pivotal role in tumor immunotherapy[39,40]. As revealed in Fig. 7f and g, the IASNDS@gel group exhibited the highest number of CD80+CD86+ cells, surpassing the Ctrl (PBS), Gel, SLIN@gel, and SDV@gel groups by 5.68, 2.67, 2.22, and 1.43 folds, respectively. Further analysis revealed a significant increase in M1 macrophages (CD80+F4/80+) and a significant decrease in M2 macrophages (CD206+F4/80+) in the IASNDS@gel group (Supplementary Fig. 21).

Further reinforcing these findings, the expression levels of cytokines associated with innate and adaptive immunity were enhanced within the GBM tissues of mice receiving IASNDS@gel treatment, and the levels of interferon γ (IFN-γ) and tumor necrosis factor α (TNF-α) were particularly elevated (Fig. 7h–j). In particular, compared to the Ctrl group, CD8+IFNγ+ T cells were significantly higher in the IASNDS@gel treatment group, suggesting a more pronounced antitumor immune response within the GBM (Supplementary Fig. 22).

Furthermore, we investigated the effect of ATP-responsive gel on the effect of activated immunization, and the non-ATP-responsive IASNDS@gel immunization was significantly less vigorous than the ATP-responsive IASNDS@gel (Supplementary Fig. 23). On further analysis, the activation and increase of immune memory cells may play a very important role (Supplementary Fig. 24). We further rechallenged mice originally bearing GBM in the right side, and found that when the mice were treated with IASNDS@ gel for 20 days and then transplanted with GBM cells to the left side, the IASNDS@ gel treatment significantly inhibited the growth of GBM, and the survival time of the mice was also prolonged (Supplementary Fig. 25).

These findings demonstrate that the introduction of the autolysing bacterium-hydrogel system into the tumor cavity of GBM mice after surgery increased the frequency of responding immune cells. This orchestrated the establishment of a comprehensive tumor

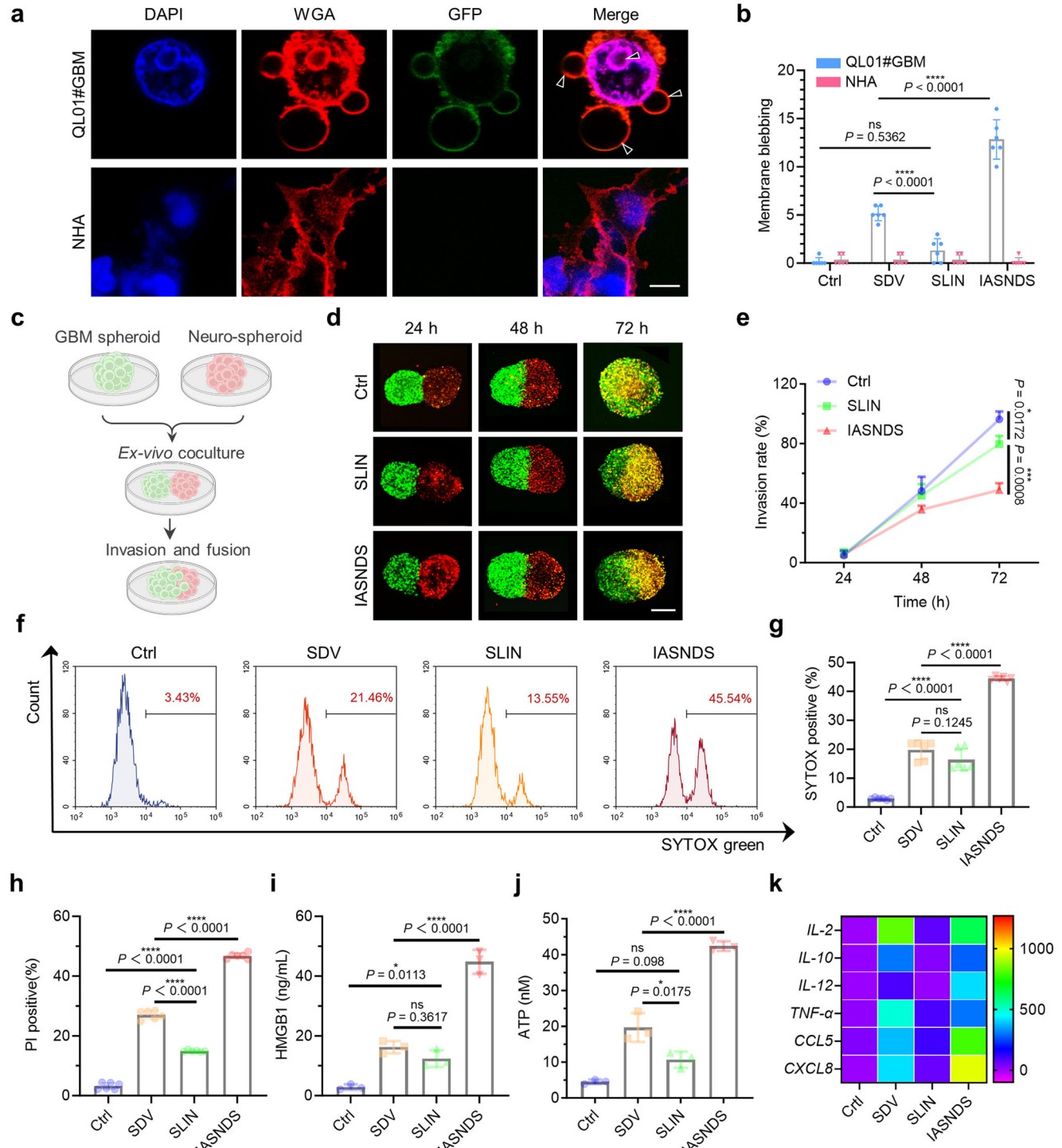

**Fig. 4 | Validation of IASNDS-induced GBM pyroptosis and ICD-related changes in vitro. a** Visualization of cellular pyroptosis changes induced by the IASNDS under confocal microscopy, showing membrane pores and vacuole formation in QL01#GBM cells (*n* = 3 independent experiments). **b** Quantification of vacuoles in cells from different treatment groups to assess pyroptosis efficiency (*n* = 6 images from three independent experiments). (exact *P* values: SLIN vs. SDV *P* = 6.28201E −05, IASNDS vs. SDV *P* = 4.22634E−06); ****P* < 0.0001. **c**–**e** 3D neurosphere and QL01#GBM tumor sphere invasion experimental design, confocal images of the extent of GBM sphere invasion over time (*n* = 3 independent experiments). **f**–**h** Flow cytometry analysis of stained cells (SYTOX, PI). (*n* = 6 independent experiments). (exact *P* values of g: Ctrl vs. SLIN *P* = 5.45783E−08, SDV vs. IASNDS *P* = 1.141E−12; exact *P* values of h: Ctrl vs. SLIN *P* = 1.24E−13, SDV vs. SLIN *P* = 7.5E−14,

SDV vs. IASNDS *P* = 1.9E−14); ****P* < 0.0001. **i, j** HMGB1 and ATP release from QL01#GBM cells after 24 hours of PBS, SDV, SLIN, and IASNDS treatment, analyzed by flow cytometry (*n* = 3 independent experiments). (exact *P* value of i: SDV vs. IASNDS *P* = 5.8384E-06; exact *P* value of j: SDV vs. IASNDS *P* = 1.38256E-05); ****P* < 0.0001. **k** PCR detection of cytokine and chemokine expression trends after 24 hours of PBS, SDV, SLIN, and IASNDS treatment of QL01#GBM cells. (*n* = 3 independent experiments). Data are presented as the mean ± S.D. The statistical comparisons in **b**, **e**, **g**–**j** were performed using one-way ANOVA with Tukey's post hoc test, with asterisks indicating significant differences (ns = no significance, **P* < 0.05, ***P* < 0.01, ****P* < 0.001, *****P* < 0.0001). **a, d** show representative images of the corresponding independent biological samples. The scale bar in **a** is 5 μm, and the scale bar in **d** is 1 mm. Source data are provided as a Source Data file.

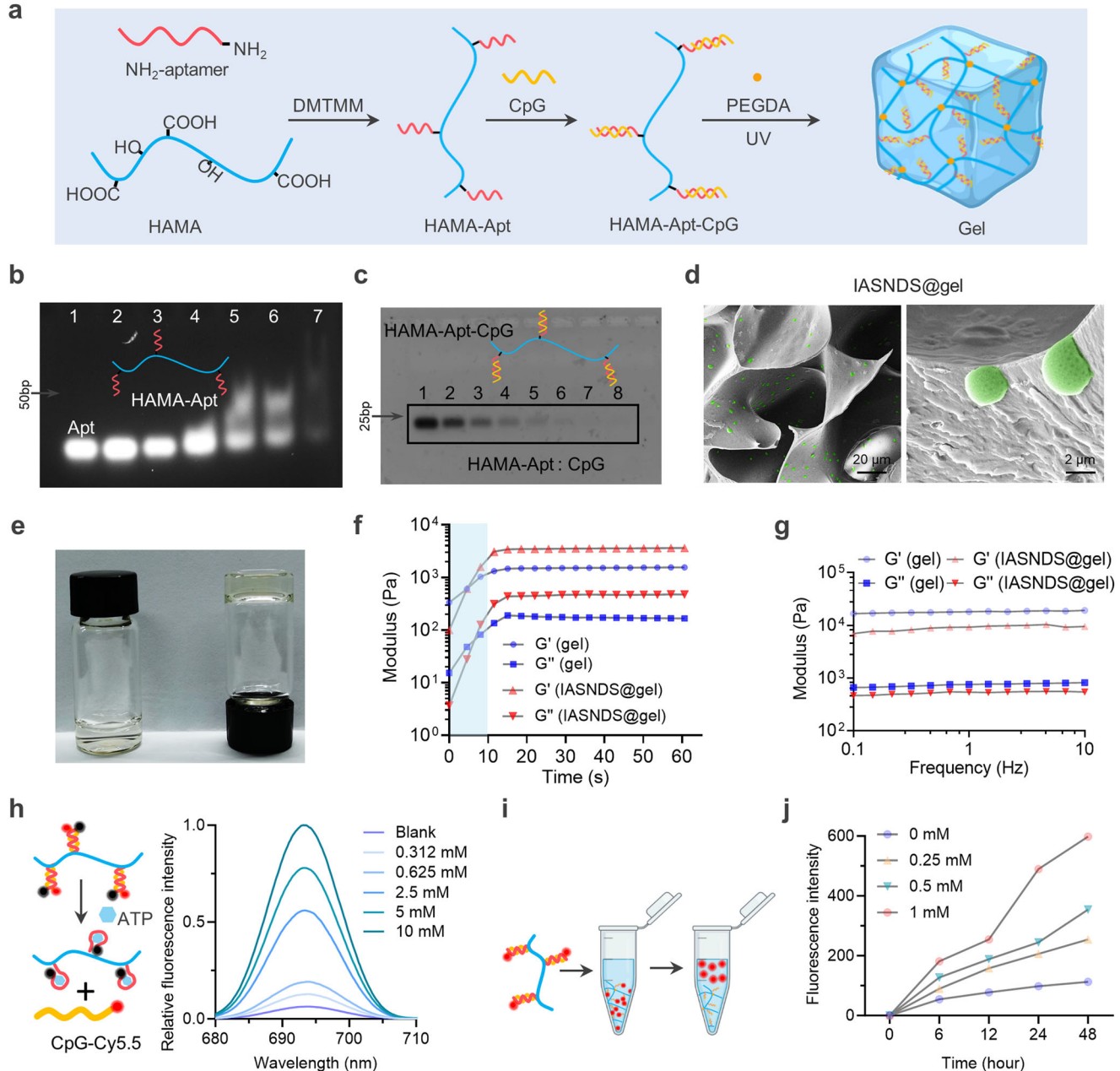

**Fig. 5 | ATP-responsive hydrogel design and characterization. a** Schematic of the ATP-responsive hydrogel used to encapsulate the IASNDS. **b** PAGE analysis of HAMA-Apt synthetic conjugates formed from reaction mixtures of ATP nucleic acid aptamer (lane 1) and Apt with HAMA at molar ratios of 100:1 (lane 2), 50:1 (lane 3), 25:1 (lane 4), 12:1 (lane 5), 6:1 (lane 6), and 1:1 (lane 7). (n = 3 independent experiments). **c** PAGE analysis of the supernatant after binding of HAMA-Apt to CpG ODN with molar ratios of HAMA-Apt to CpG ODN of 1:0 (lane 8), 1:1 (lane 7), 1:2 (lane 6), 1:4 (lane 5), 1:8 (lane 4), 1:16 (lane 3), 1:32 (lane 2) and 1:64 (lane 1). (n = 3 independent experiments). **d** Representative scanning electron microscope image of the hydrogel (n = 3 independent experiments). **e** Representative photographs of hydrogels before and after gel formation. **f** Variation in the hydrogel modulus with time, measured at an angular frequency of 1 rad s⁻¹. (n = 3 independent experiments). **g** Analysis of hydrogel variation with frequency, measured at a fixed strain of 0.5%. (n = 3 independent experiments). **h** Schematic of ATP-induced dehybridization of double-stranded BHQ3-modified Apt and Cy5.5-modified CpG structures, with fluorescence spectra showing the fluorescence recovery of Cy5.5 under ATP induction. (n = 3 independent experiments). **i** Schematic showing the release of Cy5-labeled CpG from the hydrogel. **j** Cumulative amount of CpG ODN released from the hydrogel under different concentrations of ATP. (n = 3 independent experiments). Data are presented as the mean ± S.D. The images in **b**–**d** are representative images of the corresponding independent biological samples. The scale bars in **d** are 20 μm (left) and 2 μm (right). Source data are provided as a Source Data file.

clearance system involving cytokines, immune cells, and the immune microenvironment. The synergistic response generated by this bacteriotherapy effectively inhibits the relapse of GBM.

## Discussion

Surgical debulking has emerged as the preferred therapeutic approach for individuals with GBM[41]. The brain parenchymal infiltration and rapid proliferation of GBM cells result in a relapse of the tumor within 2-3 cm from the surgical margin in approximately 90% of patients within 8 months after the operation[42–44]. Optimizing surgical resection while preserving the integrity of normal brain function is a challenging goal for neurosurgeons. Given the existing treatment strategies, complete surgical resection does not appear to be feasible, suggesting that the main avenue for enhancing the prognosis and quality of life of

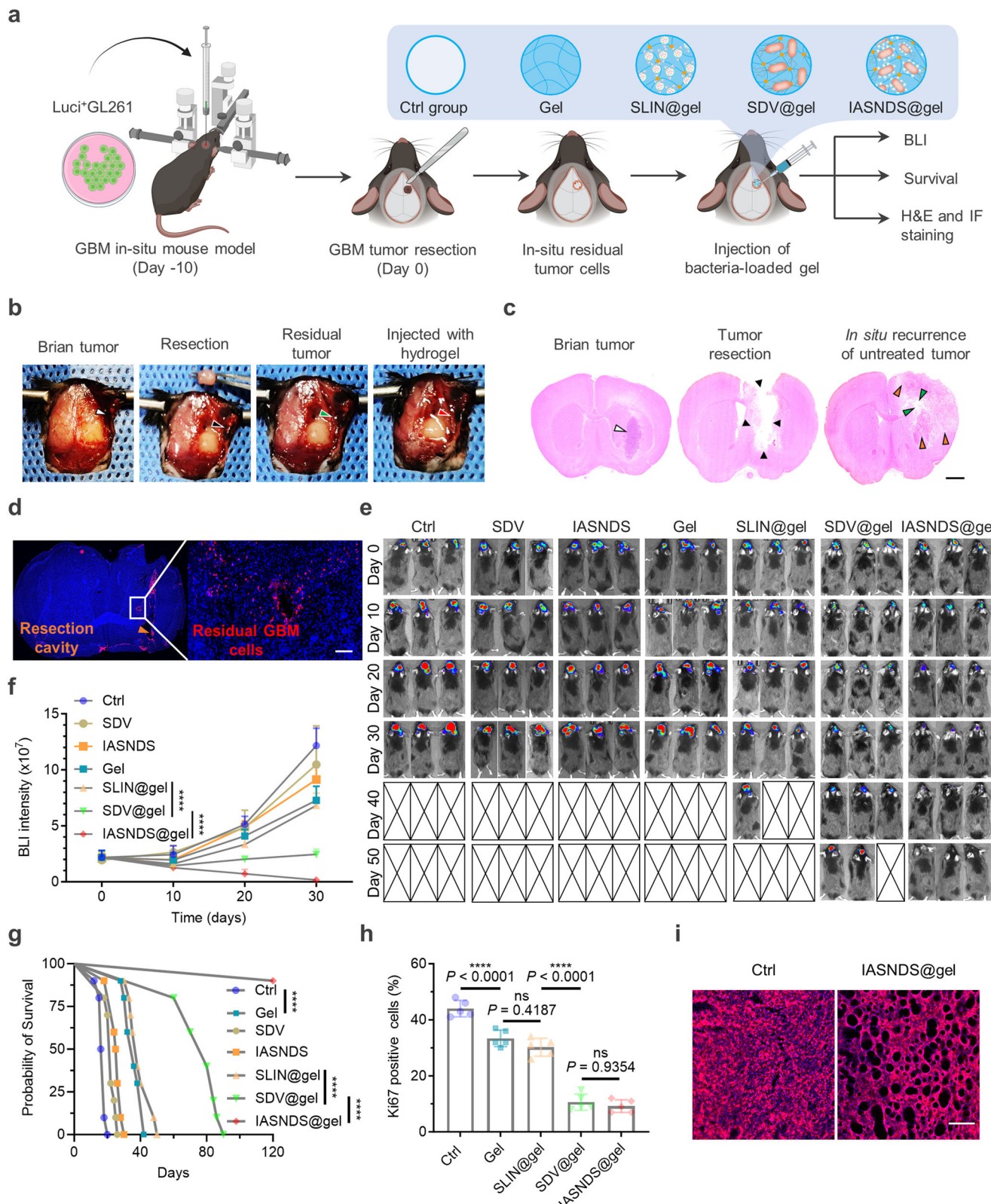

patients struggling with GBM lies in the prevention of recurrence after surgical intervention. In this study, residual GBM cells invading the tumor margins were removed by the combined application of an autolytic Salmonella vehicle and an injectable ATP-responsive hydrogel. This surgical adjuvant approach possesses the distinctive advantage of harmonizing with the contemporary treatment landscape and has the potential for broad application in the clinic, specifically

addressing the challenge of managing residual GBM cells following surgery.

Bacteriotherapy safety is a prerequisite for GBM treatment. Intracranial application of bacteriotherapy for the treatment of tumors has been investigated in less frequently, and the main issue considered is the specificity of intracranial immunity. To address this, we tethered nanoparticles inducing bacterial lysis on the surface of SDVs to initiate

**Fig. 6 | Suppression of postoperative recurrence and prolonged survival via intracavity injection of IASNDS@gel. a** Illustration of the in vivo experimental design. **b** Upon reaching day 10 after initial inoculation, tumor-bearing GL261 mice underwent surgical tumor resection, followed by hydrogel injection. Subsequently, brain tissues from the excised mice were subjected to euthanasia. H&E (**c**) and immunofluorescence (**d**) staining was carried out to visualize residual GBM cells (labeled red with a Vimentin primary antibody). ($n = 3$ independent experiments). **e** In vivo bioluminescence images were captured ($n = 3$), and **f** the resulting signal intensity was quantified ($n = 10$). (exact $P$ values: SLIN@gel vs. SDV@gel $P = 1.05777E-09$, SDV@gel vs. IASNDS@gel $P = 6.4922E-11$); ****$P < 0.0001$. **g** Mouse survival was assessed through Kaplan–Meier survival curves for each treatment group ($n = 10$). Data analysis was performed employing the log-rank (Mantel–Cox) test. (exact $P$ values: Ctrl vs. Gel $P = 2.55345E-08$, SLIN@gel vs. SDV@gel

$P = 1.60341E-08$, SDV@gel vs. IASNDS@gel $P = 3.26493E-05$); ****$P < 0.0001$. **h** The percentage of positive cells was tallied following immunohistochemical staining for Ki-67 in tumor tissues originating from various mouse groups ($n = 5$ images from three independent experiments). (exact $P$ values: Ctrl vs. Gel $P = 8.876E-05$, SLIN@gel vs. SDV@gel $P = 7.74109E-09$); ****$P < 0.0001$. **i** Visualization of alterations in mouse brain tissues was achieved by applying WGA staining (red), enabling observation of tumor cell dynamics across the different treatment groups in mice ($n = 3$ independent experiments). Data are presented as the mean ± S.D. The statistical comparisons in **f** and **h** were performed using two-way ANOVA and one-way ANOVA by Tukey's post hoc test, with asterisks indicating significant differences (***$P < 0.001$). The images in **c**, **d**, and **i** are representative images of the corresponding independent biological samples. The scale bar in **c** is 1.5 mm, and the scale bar in **d** and **i** is 50 μm. Source data are provided as a Source Data file.

the self-destructive process of the bacteria within the GBM cells. We experimentally verified that SDVs can target the hypoxic microenvironment of GBM and can colonize within the GBM, while those in peripheral organs are easily eliminated (Supplementary Fig. 28). Localized application of SDVs in the operative cavity also did not cause infection, thus demonstrating the superior safety of this therapeutic modality (Supplementary Table 1).

The synergistic interplay between innate and adaptive immunity strengthens the immune response against tumors, effectively preventing the proliferation and dissemination of GBM cells and enhancing therapeutic outcomes. The SDV designed here possesses the capability to colonize, propagate, and release therapeutic components in GBM cells, thereby exerting direct or indirect cytotoxic effects on tumor cells. Additionally, bacteria can engage with innate immune receptors, including Toll-like receptors and others, through a diverse array of molecules that are either present on their surface or secreted; these molecules include lipopolysaccharides, lipoproteins, flagellin, chemokines, and others (Supplementary Fig. 15). Innate immune signaling induces the secretion of substantial amounts of cytokines and chemokines, including TNF-α, IL-2, IL-10, IL-12, and others (Fig. 4k). Furthermore, bacterial lysates are also involved in the activation of adaptive immune cells through antigen cross-presentation (Fig. 7h–j). We demonstrated that the strategy of activating innate and adaptive immunity via bacteriotherapy is a practical way to reverse the immunosuppressive microenvironment and promote the formation of immune memory to prevent tumor recurrence.

In summary, the development of bacterial therapies using a surface tethering strategy has the potential to enhance the efficacy of immunotherapy for GBM while improving the safety of bacteriotherapy. The activation of both innate and adaptive immune responses to prevent GBM relapse after surgery, facilitated by the delivery of an autocleaving Salmonella therapeutic system via an immune-stimulating hydrogel, represents a high-efficiency and broad-spectrum immunotherapeutic strategy. Our work reports a local bacteriotherapy that stimulates anticancer immunity and can be widely used in patients with malignancies with a high risk of recurrence.

## Methods
### Animals
All procedures were approved by the Research Ethics Committee of Shandong University and the Ethics Committee of Qilu Hospital (Shandong, China), in compliance with all relevant ethical regulations. According to the ethics committee, the size of the tumor must not exceed 10% of the animal's body weight, while the animal must not lose more than 20% of its body weight during the research. The maximal tumor burden was not exceeded in this study. Sex was not taken into account in the study design. The mice were kept in a barrier environment with a constant temperature of 24°C and a relative humidity of 50%. The mice were maintained under a 12-hour light and 12-hour dark cycle.

## Materials
All reagents and solvents (analytical grade) were used as supplied by commercial sources unless otherwise indicated. Hyaluronic acid methacryloyl (HAMA) was purchased from Engineering for Life (Suzhou, China). All aptamers and CpG ODNs were synthesized by Synbio Technologies (Suzhou, China). The GSDMD-linker-BASP1 lentiviral plasmid, called pcSLenti-EF1-EGFP-P2A-Puro-CMV-GSDMD-linker-BASP1-3xFLAG-WPRE, was constructed, and packaged by OBiO Technology Corp., Ltd. (Shanghai, China). VNP20009 was purchased from Biosci Technology (Hangzhou, China). Fetal bovine serum (FBS), neurobasal medium, B-27 serum-free supplement, epidermal growth factor (EGF), basic fibroblast growth factor (bFGF), Dulbecco's modified Eagle's medium (DMEM), penicillin and trypsin-EDTA were obtained from Thermo Fisher Scientific Inc. (Shanghai, China). Cell culture freezing medium and enhanced chemiluminescence were purchased from New Cell & Molecular Biotech Co. Ltd. (Suzhou, China). Wheat germ agglutinin was purchased from Invitrogen. DAPI, BCA protein assay kits, RIPA lysis buffer, and ATP assay kits were all obtained from Beyotime Biotech Inc. (Shanghai, China). Horseradish peroxidase-conjugated secondary antibody was purchased from ZSGB-BIO (Beijing, China). A PCR array was purchased from Wcgene Biotech (Shanghai, China). C57BL/6 mice (male, 6 weeks) were purchased from GemPharmatech Co., Ltd. (Nanjing, China). ELISA kits for TNF-α and IFN-γ were purchased from PeproTech, Inc. (Cranbury, NJ, USA). The antibodies used in this study were summarized as follows (company, Catalog number, Clone name, dilutions): CD3-PerCP-Cy5.5 (BioLegend, Catalog number: 100218; Clone name: 17A2; 1:20 dilution); CD4-PE (BioLegend, Catalog number: 100408; Clone name: GK1.5; 1:100 dilution); CD8-FITC (BioLegend, Catalog number: 100706; Clone name: 53-6.7; 1:50 dilution); Granzyme B-Alexa Fluor 647 (BioLegend, Catalog number: 372220; Clone name: QA16A02; 1:20 dilution); IFNγ-PE (Invitrogen, Catalog number: 12-7319-41; Clone name: 4S.B3; 1:20 dilution); CD80-PE (BioLegend, Catalog number: 104708; Clone name: 16-10A1; 1:40 dilution); CD86-APC (BioLegend, Catalog number: 105012; Clone name: GL-1; 1:80 dilution); CD206-APC (BioLegend, Catalog number:141708; Clone name: C068C2; 1:40 dilution); F4/80-FITC (BioLegend, Catalog number:123108; Clone name: BM8; 1:200 dilution); β-Actin (Cell Signaling Technology, Catalog number: 4970 S; Clone name: 13E5; 1:1000 dilution); Calnexin (Cell Signaling Technology, Catalog number: 2679 S; Clone name: C5C9; 1:1000 dilution); Gasdermin D (Cell Signaling Technology, Catalog number: 39754 S; Clone name: E9S1X; 1:1000 dilution); GSDMD-N (Cell Signaling Technology, Catalog number: 36425S; Clone name: Asp275; 1:1000 dilution); Cleaved Caspase-1 (Cell Signaling Technology, Catalog number: 4199 T; Clone name: Asp297; 1:1000 dilution); TSG101 (Abcam, Catalog number: ab125011; Clone name: EPR7130; 1:1000 dilution). The sequences of all the DNA, primers, and CpG sites can be found in Supplementary Table 2 and Supplementary Table 3.

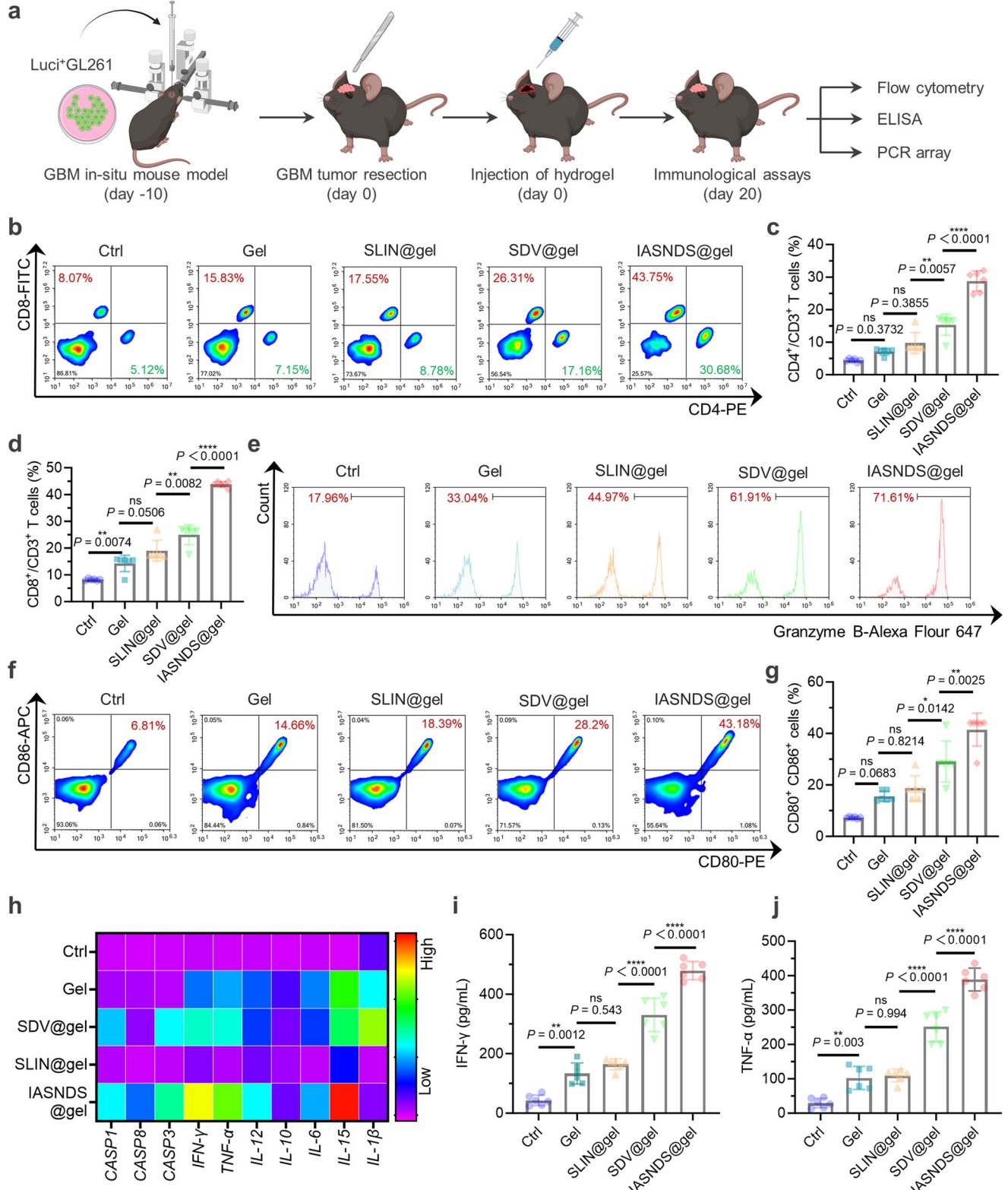

## Cell lines and cell culture

Sex or gender were not taken into account in the study design. The mouse GBM cell line GL261 was acquired from the Cell Bank of the Chinese Academy of Sciences (Shanghai, China), while mega-karyocytes (MEG01) were obtained from the American Type Culture Collection. Surgically isolated QL01#GBM primary human GBM cells were obtained from the Department of Neurosurgery of Qilu Hospital, Shandong University, with informed consent from the patient

and approval from the Ethics Committee of Qilu Hospital. GL261 cells were cultured in DMEM supplemented with 10% FBS in a humidified incubator with 5% $CO_2$ at 37°C. MEG01 cells were cultured in Roswell Park Memorial Institute (RPMI) media supplemented with 10% FBS and 2 mM glutamine in a humidified incubator at 37 °C supplied with 5% $CO_2$. QL01#GBM cells were cultured in Neurobasal Medium supplemented with 2% B-27 Serum-Free Supplement, 20 ng/ mL EGF, and 10 ng/mL bFGF in a humidified incubator with 5% $CO_2$ at

**Fig. 7 | Immunomodulatory efficacy of the Salmonella-loaded hydrogel system for mobilizing immunity against tumors via intracavity injection. a** Schematic of the experimental design for assessing immunomodulatory efficacy. **b** Flow cytometry results showing CD3$^+$ cells in brain tissues after GBM removal in each treatment group. **c** Quantitative analysis of CD3$^+$CD4$^+$ cytotoxic T cells ($n = 6$ independent experiments). (exact $P$ value: SDV@gel vs. IASNDS@gel $P = 1.57104E$-08); ****$P < 0.0001$. **d** Quantitative analysis of CD3$^+$CD8$^+$ cytotoxic T cells ($n = 6$ independent experiments). (exact $P$ value: SDV@gel vs. IASNDS@gel $P = 1.32267E$-10); ****$P < 0.0001$. **e** Flow cytometry results indicating CD3$^+$CD8$^+$Granzyme B$^+$ T cells in GBM tissues from different treatment groups. **f, g** Flow cytometry analysis and statistical analysis of CD80$^+$CD86$^+$ cells in GBM tissues after different treatments ($n = 6$ independent experiments). **h** Heatmap displaying the expression profiles of pyroptosis-related proteins, cytokines, and chemokines in brain tumor tissues. ($n = 3$ independent experiments). **i, j** Statistical analysis of IFN-γ expression and TNF-α expression in GBM tissues under different treatment conditions ($n = 6$ independent experiments). (exact $P$ values of i: SLIN@gel vs. SDV@gel $P = 1.22239E$−07, SDV@gel vs. IASNDS@gel $P = 8.7998E$−07; exact $P$ values of j: SLIN@gel vs. SDV@gel $P = 2.02309E$−07; SDV@gel vs. IASNDS@gel $P = 3.4876E$−07); ****$P < 0.0001$. Data are presented as the mean ± S.D. The statistical comparisons in **c, d, g, i** and **j** were performed using one-way ANOVA with Tukey's post hoc test, with asterisks indicating significant differences (ns = no significance, *$P < 0.05$, **$P < 0.01$, ***$P < 0.001$, ****$P < 0.0001$). Source data are provided as a Source Data file.

37°C. Fluorescein-labeled GL261 cell lines were identified using an in vivo imaging system to assess luminescence intensity.

## Preparation and characterization of the IASNDS

A lentiviral plasmid was designed for overexpression of GSDMD on the surface of exosomes by combining the gene for the exosome "scaffolding" protein BASP1 with the GSDMD gene. The plasmid was then transfected with the envelope plasmid into tool cells to construct the lentivirus. The lentivirus was transfected into human megakaryocytes (MEG01) to further obtain exosomes stably expressing GSDMD on the surface. Cell cultures were collected after starvation and centrifuged at 5000 x $g$ for 20 min and 12,000 x $g$ for 30 min to remove the sediment. The supernatant was transferred to another tube, filtered through a 0.22 μm filter, and then ultra-centrifuged at 100,000 x $g$ for 70 min (Beckman Optima L-100 XP, Beckman Coulter), after which the precipitated exosomes were resuspended in PBS.

To load the L-arabinose into the exosome interior, we mixed 10$^9$ exosomes and 1 mM L-arabinose in an electroporation buffer. SLINs were synthesized on the basis of L-arabinose loaded into exosomes using a single 4 mm cuvette and a Lonza Nucleoector 2B system, followed by the application of cold PBS solution to dilute the exosomes and centrifugation at 100,000 x $g$ for 70 min to remove the unbound drug. To achieve the modification of DBCO on the surface of SLINs, we resuspended the SLINs in PBS (0.5 mg mL$^{-1}$), added 3 mM DBCO-sulfo-NHS, and finally removed unbound DBCO-sulfo-NHS by ultrafiltration. The DBCO-SLINs and azide-SDVs were then coupled using copper-free click chemistry, and 2 h of incubation yielded the IASNDS.

To observe the morphology of the synthesized nanodrugs by TEM, we added them dropwise on 200 mesh Formvar- and carbon-coated copper grids (Ted Pella) and left them for 1 min. The excess drug solution was then rinsed off with water, and the grids were stained with 1% uranyl acetate solution for 30 seconds. After staining, the grids were air-dried, and images of the negatively stained samples were captured using an 80-kV transmission electron microscope (TEM, JEOL JEM-1400, Japan). Characterization of the IASNDS was then performed.

## Cell invasion assays

To assess 3D tumor sphere invasion, QL01#GBM cells (3 × 10^3/well) were seeded into low-adherence 96-well plates. After sphere formation, the invasion gel (50 μL/well; R&D Systems; 3500−096-03; Minneapolis, MN, USA) was introduced, followed by a 72-hour incubation period. Throughout this timeframe, images were captured at specific intervals to analyze the invasion potential of the engineered tumor cells. To evaluate GBM cell invasion within an in vitro brain organoid context, 18-day rat embryonic brain organoids were cultivated. GFP-transfected GBM cells capable of forming spheres were cocultured with fully developed brain organoids for a duration of 72 hours. High-resolution images of these cocultures were obtained using confocal microscopy (Leica TCS Sp8, Wetzlar, Germany) to assess invasion capabilities.

## Western blot analysis

Protein was extracted from cells and exosomes using RIPA lysis buffer containing PMSF protease inhibitor. BCA analysis was performed to determine the protein concentration. Then, the protein samples with added protein loading buffer were boiled for 10 min in a 100 °C metal bath and stored at −20 °C for further use. To perform the Western blot analysis, equal amounts of protein extracts were separated by 10% SDS–PAGE and transferred to PVDF membranes (Merck Millipore, Billerica, MA, USA). Blocking was performed with skim milk for 1 hour at room temperature, followed by overnight incubation at 4 °C with the primary antibody. The membranes were then incubated for 1 hour at room temperature with horseradish peroxidase-conjugated secondary antibody diluted in antibody dilution buffer. Finally, proteins on the membrane were visualized by chemiluminescence (Bio-Rad, Hercules, CA, USA) according to the manufacturer's instructions.

## Establishment of an animal model for the antitumor efficacy study

The intracranial GBM mouse model was established with C57BL/6J mice (6-8 weeks of age). Luciferase-labeled GL261 cells were used for intracranial xenograft experiments by injecting 1 × 10^6 cells diluted in 10 μL PBS into the right frontal lobe of each mouse at the coordinates of 1 mm anterior and 2.5 mm lateral to the bregma, with a depth of 2 mm, and the corresponding area of the skull was removed. To establish the GBM resection mouse model, the GBM tumor was surgically excised from tumor-bearing mice under magnification on day 10 after inoculation, and then 30 μL of hydrogels with different formulations were injected into the tumor excision cavity of each mouse. Tumors were also monitored by BLI using the IVIS Spectrum system (Perkin-Elmer; MA, USA). Euthanasia was performed on mice that showed severe hunching, apathy, decreased activity, leg flopping, and significant weight loss.

Mice were immobilized before tail vein injection. The tail vein was rinsed with ethanol to induce congestion and dilation in preparation for injection. The injection started from the end of the tail vein, choosing an entry point close to the thumbnail and placing the needle at an angle of approximately 30° to the vessel. The needle is gently inserted into the skin with the beveled side facing up. After insertion, align the needle parallel to the vessel, keeping the hand steady to prevent fluid leakage.

## Stability assay for SLINs

We restricted SLINs to dialysis cassettes with a diameter of 10 nm co-incubated them with 1 ×10^7 CFUs of SDVs, and then detected changes in the diameter of SLINs at different times. The dialysis cassettes selectively allowed SDV metabolites and secreted proteins to enter the culture fluid. At the end of the culture, we used dynamic light

scattering (DLS) to detect the particle size of SLINs and then statistically analyzed the observed changes.

## ELISA

ELISA kits were used to detect the secretion of cytokines in tumor tissues through a double antibody sandwich enzyme-linked immunosorbent assay technique. The wells of ELISA plates were coated with captured antibodies overnight at room temperature, followed by incubation with blocking buffer for 1 hour. Standards and samples were then added to each well in triplicate in equal amounts. After incubation for 2 hours at room temperature, 100 μL of detection antibody was added to each well. The plate was reincubated for 2 hours and then blotted and washed 4 times. Then, 100 μL of diluted streptavidin-HRP conjugate was added to each well. After incubation for 30 minutes at room temperature, the plate was washed. Then, 100 μL of substrate solution was added to each well and incubated for 20 minutes at room temperature. Finally, 100 μL of 1 M HCl stop solution was added. Color development was monitored with a microplate reader (Bio-Rad, CA, USA) at 450 nm with wavelength correction set at 620 nm.

## Flow cytometry analysis

Each group of mice was euthanized on day 20 post-treatment. Tumor tissue was collected, cut into small pieces, and then ground in a glass grinder with cold PBS buffer to obtain a cell suspension. The suspension was centrifuged at 3000 rpm for 5 minutes at 4°C, and the precipitate was retained. Cells were resuspended in PBS and then filtered through a 70 μm nylon cell filter, counted, and analyzed by flow cytometry. CD16/CD32 (1:1500 dilution) was added to block the cells for 15 minutes to prevent nonspecific antibody binding. Anti-CD3-PerCP-Cy5.5 (1:20 dilution), anti-CD4-PE (1:100 dilution), and anti-CD8-FITC (1:50 dilution) were added to stain the cells for $CD3^+CD4^+$ and $CD3^+CD8^+$ T-cell analysis. $CD3^+$ cells were stained with anti-CD8-FITC (1:50 dilution), fixed, permeabilized, and stained with anti-Granzyme B-Alexa Fluor 647 (1:20 dilution) in accordance with the manufacturer's instructions. For $CD80^+CD86^+$ antigen-presenting cell analysis, cells were stained with anti-CD80-PE (1:40 dilution) and anti-CD86-APC (1:80 dilution). The gating strategy for the T-cell analysis is presented in Supplementary Fig. 19.

## Statistics and reproducibility

All data were analyzed using GraphPad Prism 9 software and are presented as the mean ± standard error of the mean (SEM). Two-way analysis of variance (ANOVA), the log-rank test, and unpaired two-tailed $t$-tests were used to calculate $P$ values for different experimental purposes. The data were tested for normality and equal variance using a $t$-test, and the results showed that the data were normally distributed. To assess survival differences between groups, Kaplan–Meier survival curves were compared using the log-rank test. The methods for data collection and analysis were predetermined before the start of the experiments. Unless otherwise specified, all experiments were repeated at least three times. In addition, two-sided tests were used for all tests. Significant differences are indicated as $*P < 0.05$, $**P < 0.01$, $***P < 0.001$ and $****P < 0.0001$.

## Reporting summary

Further information on research design is available in the Nature Portfolio Reporting Summary linked to this article.

## Data availability

All data generated from this study are available within the Article, Supplementary Information or Source Data file. Source data are provided with this paper.

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

## Acknowledgements

This work was supported by the National Natural Science Foundation of China (82111530202 to S.N., 82172740 to S.N., 82303810 to Yulin Zhang, 22202119 to H.G.), the Natural Science Foundation of Shandong Province (ZR2022ZD17 to S.N., ZR2023ZD15 to L.S., ZR2023QH224 to Yulin Zhang), the China Postdoctoral Science Foundation (2022M721967 to Yulin Zhang), the Innovation Project of Jinan Science and Technology Bureau (2021GXRC065 to S.N.), the Shandong Postdoctoral Innovative Talents Program (SDBX2022008 to Yulin Zhang), the Research Project of Jinan Microecological Biomedicine Shandong Laboratory (JNL-2023004C to S.N., JNL-2023013D to Yulin Zhang) and Taishan Scholar Program of Shandong Province (NO.tstp20230656 to S.N., NO.tsqnz20221165 to Yulin Zhang, tsqn202306358 to H.G.). We acknowledge theoretical and technical support from Laboratory Animal Center, Qilu Hospital of Shandong University. We thank the Translational Medicine Core Facility of Shandong University for consultations and the use of instruments that supported this work. Figures 1b, 3a, b, 4c, 5a, 5i, 6a, 7a and Supplementary Fig. 25a, created with BioRender.com, released under a Creative Commons Attribution-NonCommercial-NoDerivs 4.0 International license.

## Author contributions

Yulin Zhang, K.X., and Z.Fu. conceived and designed the project. Yuying Zhang, Z. Fang, and Yi Zhang participated in cell culture and membrane acquisition. Y.D., M.W., and J.S. produced and characterized the nanocarrier. Z. Fang, M.W., X.H., B.C. and K.X. participated in the in vivo tumor formation and antitumor experiments. K.X. and Z.Fang contributed to Western blotting. Yulin Zhang, Z.Fu., Yuying Zhang, F.F. and Yi Zhang contributed to the ELISA experiments and flow cytometry analysis. C.C. and H.G. performed in vivo imaging and analysis. X.L., L.S., X.J., and S.N. analyzed and interpreted the data in this study. Yulin Zhang, K.X., and Z.Fu. drafted the manuscript, which was approved by all coauthors.

## Competing interests

The authors declare no competing interests.

## Additional information

[1]Department of Neurosurgery, Qilu Hospital, Cheeloo College of Medicine, Shandong University, 107 Wenhua Xi Road, Jinan 250012 Shandong, China. [2]Institute of Brain and Brain-Inspired Science,, Shandong University, 107 Wenhua Xi Road, Jinan 250012 Shandong, China. [3]Key Laboratory of Chemical Biology (Ministry of Education), Department of Pharmaceutics, School of Pharmaceutical Sciences, Cheeloo College of Medicine, Shandong University, 44 Wenhua Xi Road, Jinan 250012 Shandong, China. [4]Department of Pediatrics, Qilu hospital, Cheeloo College of Medicine, Shandong University, 107 Wenhua Xi Road, Jinan 250012 Shandong, China. [5]Department of Obstetrics, The Second Hospital, Cheeloo College of Medicine, Shandong University, No. 247 Beiyuan Road, Jinan 250033 Shandong, China. [6]Department of Radiation Oncology, Qilu Hospital affiliated to Shandong University, 107 Wenhua Xi Road, Jinan 250012 Shandong, China. [7]Department of Endocrinology, Qilu Hospital, Cheeloo College of Medicine, Shandong University, 107 Wenhua Xi Road, Jinan, Shandong 250012, China. [8]Jinan Microecological Biomedicine Shandong Laboratory, Jinan 250117 Shandong, China. [9]Shandong Key Laboratory of Brain Function Remodeling, Jinan 250117 Shandong, China. [10]These authors contributed equally: Yulin Zhang, Kaiyan Xi, Zhipeng Fu. [11]These authors jointly supervised this work: Chen Chen, Xinyi Jiang, Shilei Ni. ✉e-mail: chenchen27@sdu.edu.cn; xinyijiang@sdu.edu.cn; nishilei@sdu.edu.cn

