## [Peer Review File · Nature Communications]

Stimulation of Tumoricidal Immunity via Bacteriotherapy Inhibits Glioblastoma RelapseREVIEWER COMMENTS

Reviewer #1 (Remarks to the Author): with expertise in bacteria-based cancer (immuno)therapy

This manuscript entitled “Stimulation of Tumoricidal Immunity via Bacteriotherapy Inhibits Glioblastoma Relapse” describes a cavity-injectable bacterium-hydrogel superstructure that targets GBM satellites around the cavity, triggers GBM pyroptosis, and initiates innate and adaptive immune responses, thereby preventing postoperative GBM relapse. This work reports a local bacteriotherapy for stimulating anticancer immunity, particularly for malignancies with a high risk of recurrence. However, considering the innovation and the issues mentioned below, this manuscript does not meet the publication level of Nature Communications.

Major issues:

1. It would be better to reorganize the abstract to make it clearer.
2. Figure 1f appears to show that the application of SDVs makes brain tissue more porous. The authors should explain the reason behind this observation.
3. In Figure 2i, the authors did not consider the interference caused by components such as nanosized extracellular vesicles secreted by Salmonella, which could affect the size measurement of SLINs. The authors must demonstrate that the size measured is indeed for SLINs and not a mixture containing Salmonella vesicles or other bacterial components.
4. It was claimed that ATP-responsive hydrogels may promote long-term immune cell activation. To make the conclusion more solid, there should be a strict non-ATP-responsive hydrogel as a control.
5. The reviewer suggests that mixing with the hydrogel may limit the movement and diffusion of bacteria. The authors should first test whether the hydrogel itself limits the bacteria from fully exerting their cell invading effects. In animal experiments, they could include a control group where the hydrogel was injected and allowed to form solidification, and then followed by local administration of IASNDS in PBS.
6. The in vitro release of CpG ODN was investigated by hydrogels in response to ATP. How is the ATP responsiveness of hydrogels in vivo? This bears significant relevance for achieving effective delivery of IASNDS into the surgical cavity.

7. Does the UV-curing process employed in hydrogel formation have any impact on the stability and activity of the encapsulated IASNDS?
8. It is hard to see bacteria in IASNDS@gel by the SEM image in Figure 5d. High quality images are needed and bacteria inside should be pointed out clearly.
9. In Figure 7, the gating strategy should be displayed. For example, the authors should include where the percent of cells of interest are obtained from (i.e. T cells from total viable cells or total CD45+ cells).
10. It is confusing that the authors evaluate the change of CD80+CD86+ T cells in Figure 7f, g. Please clarify.
11. In Figure 7h, the expression profiles display the highest increment in IL-15 expression in IASNDS@gel group. The authors should affirm the protein level of IL-15 expression and discuss the underlying mechanism.
12. Normal cells and bacteria should be set as a control for the imaging of cellular pyroptosis and bacterial lysis, respectively.
13. In addition to visual imaging of bacterial lysis (Fig 3d), alternative methods are recommended to be used for further confirming the occurrence of bacterial lysis.
14. To further evaluate the immunomodulatory efficacy of the Salmonella-loaded hydrogel system against tumors, the authors are recommended to assess IFN γ +CD8+ T cells, M1 macrophages, and M2 macrophages.

Minor issues:

1. The full name is required when the abbreviation first appears, it can help readers to follow and understand. For example, what is the FNR, ICD, CFU and H&E in the paper? Their full names should be added.
2. The key primers used in this work should be provided in supplementary information. There is nd9 and nd10 in the manuscript, but no corresponding information available in SI. Please check it carefully.
3. The descriptions should be consistent throughout the entire manuscript. For example, "h" or "hours", " 1×10^8 " or " 1×10^8 ", and many others... Please check thoroughly.
4. There are a few typos throughout the manuscript. Please doublecheck.

Reviewer #2 (Remarks to the Author): with expertise in GBM, immunotherapy, engineering

This study introduces a solution for GBM treatment: a bacterium-hydrogel structure administered into the cavity after surgery. This system targets and triggers the death of GBM cells, prompting both innate and adaptive immune responses that deter tumor recurrence. Specifically, this approach utilizes a modified Salmonella delivery vehicle (SDV), derived from weakened Salmonella typhimurium (VNP20009), to seek out and attack GBM cells. Additionally, Salmonella lysis-inducing nanocapsules (SLINs) attached to the SDV stimulate an innate anti-tumor response by releasing bacterial components within the cells. Administered through intracavitary injection using an ATP-responsive hydrogel, this method recruits immune cells, aiding in antigen presentation and initiating an adaptive immune response. Although the therapeutic strategy is remarkable, this study lacks proper control groups in animal studies and assessment of therapeutic efficacy in different GBM models that are highly immunosuppressive. The other suggestions for further improving the study are outlined below:

1. The authors conducted immunohistochemistry (IHC) (Figure 1f) and immunofluorescence (IF) (Figure 1g) analyses to illustrate the distribution/accumulation of VNP and the gene-edited VNP variant (referred to as SDV) within the tumor of an orthotopic mouse glioma model subsequent to intravenous injection. However, the IHC (Figure 1f) did not distinctly display the distribution of either VNP or SDV within the tumor. Furthermore, the IF (Figure 1g) failed to definitively establish the accumulation of SDV-GFP within the tumor compared to the normal brain tissue. It is recommended that the authors stain the tumor cells and conduct colocalization studies to ascertain the association between tumor cells and VNP/SDV.
2. Notably, the distribution of VNP and SDV in other organs subsequent to intravenous injection was not addressed. The authors should isolate tissue suspensions from these organs to assess the potential dissemination of VNP and SDV in vitro.
3. In Figure 1h, the authors employed 500 μ M L-arabinose to induce the lysis of LysE+ SDV-GFP; however, this concentration may be excessive. It is suggested to use a significantly

lower concentration to evaluate the sensitivity of LysE+ SDV-GFP to L-arabinose accurately.

4. The construction details of the plasmids utilized in the study lack specificity. The authors are recommended to provide detailed sequences and relevant information. Additionally, the steps involved in the infection/transduction of the plasmids into the cells/bacteria are not clearly delineated.

5. The SLIN nanoparticles encapsulating L-arabinose exhibit high stability in vitro. It is essential to elucidate the mechanism triggering the release of L-arabinose from the SLIN nanoparticles in vivo.

6. The study indicates that the endosomal sorting complexes required for transport (ESCRT) III-mediated cell membrane repair significantly mitigates tumor cell pyroptosis by repairing and subsequently eliminating gasdermin pores. However, the current system does not impede the repair of the cell membrane mediated by ESCRT III.

7. Regarding the encapsulation of SDV or IASNDS in CpG-functionalized gel, the individual functionalities of SDV or IASNDS alone remain unclear, particularly in the absence of these agents encapsulated in a normal gel (CpG-unfunctionalized gel).

8. IASNDS@gel demonstrates remarkable therapeutic efficacy and extended survival in the GL261 resection model. However, the performance of intracavity injection of IASNDS@gel in unresected GBM models remains undisclosed.

9. A more comprehensive analysis of the immunoprofiling of other immune cell populations is recommended to provide a more holistic understanding.

Reviewer #3 (Remarks to the Author): with expertise in bacteria-based cancer (immuno)therapy

This study aims to demonstrate the efficacy of a Salmonella-based system in preventing the recurrence of glioblastoma post-surgery. The authors constructed an attenuated VNP20009

strain of *Salmonella* with Δ msbB, Δ purl, and Δ xyl deletions to carry two types of plasmids. The first plasmid contains *InvA*, controlled by the hypoxic promoter *FNR*, enabling bacterial invasion into target cells in the hypoxic tumor. The second plasmid contains *LysE*, controlled by the pBAD promoter, to lyse the bacteria and release their contents. This engineered *Salmonella* is referred to as the "Salmonella delivery vehicle" (SDV). Additionally, the authors created exosomes, termed "Salmonella lysis-inducing nanocapsules" (SLIN), derived from megakaryocytes. These exosomes carry a fusion protein of exosome-localized *BASP1* and the pore-forming protein essential for cellular pyroptosis, *gasdermin D* (*GSDMD*). The exosomes are encapsulated with L-arabinose and attached to the *Salmonella* membrane. Finally, this construct is combined with an immune-activating hydrogel, which couples CpG ODN with a nucleic acid aptamer and cross-links them using UV irradiation to retain the therapeutic factors locally. The authors demonstrate that the system is functional and effective in preventing the invasion and recurrence of brain tumors after surgery.

Although this study was relatively well-designed and the design and concept appeared to be intriguing, this manuscript is not comprehensibly written. Many parts are omitted and dispersed in results, figure legends, and materials & methods. Moreover, specific issues should be discussed in the Discussion section.

Major comments:

1. In line 67, "NLRs located on the surface of phagocytes" is incorrect. NLRs is located in the cytosol.
2. This paper is not written in a comprehensible manner. Explanations for many parts are omitted. For instance, there should be an explanation in the results section regarding the purpose of using *BASP1* whose full name is not provided at the appropriate place. Additionally, the sentences in lines 112-115 are difficult to understand. The meaning of the 'unlocked' structural configuration is unclear, and the logic claiming that it guarantees safety is also questionable. Moreover, there is mention of adding L-arabinose to exosomes. Meaning behind 'adding it to exosomes' is not clear.
3. The paragraph between 118-128 should be rewritten because the explanation of Supplementary figure 1 is not enough and it is not clear how hypoxic and normoxic conditions are made.

4. Figure 1g demonstrated SDV successfully colonize brain tumors, while VNP20009 failed. SDV is a transformed strain of VNP20009 by *invA* and *LysE*. What is the reason of the difference between two strains. *InvA* and *LysE* expression affects the targeting capability?
5. SDV and SLINs have gained functions by engineering, but the gain of function was not fully evaluated and verified. For example, *InvA* was employed to better invade the cancer cells, but the function of *InvA* was not evaluated. The ratio of invasion of SDV with or without *InvA* should be quantified. Additionally, while *InvA* allows Salmonella to invade epithelial cells, it is important to note that GBM cells, such as GL261, are not epithelial cells. They are glial cells. Does *InvA* work with glial cells? In addition, P-selectin was employed in SLIN for binding to CD44 of GBM cells, but binding of P-selectin and CD44 was not verified. Furthermore, the expression level of CD44 in GBM should be presented as compared to normal tissue.
6. The mechanism of conjugating exosomes (SLIN) to Salmonella VNP20009 (SDV) is not explained.
7. Figure 1 needs to convey more information. It should explain where GFP is encoded, as well as what is present on the surface and inside the exosomes. In other words, Fig 2f should be demonstrated in Figure 1b.
8. In this study, gasdermin family proteins, the membrane perforin gasdermin D (GSDMD) was used to trigger cellular pyroptosis, which releases tumor antigens, cytokines, and chemokines that significantly activate antitumor immune responses (line 183). Although Western blot analysis for cellular GSDMD expression was shown in supplementary Fig. 5, therapeutic mechanism using this protein was not explained clearly. Additional experiments or explanations are needed, including comparing treatment effects depending on whether GSDMD is expressed or not.
9. Each step such as the release of L-arabinose from exosomes after Salmonella invades GBM cells, activating the pBAD promoter, and the expression of *LysE* under these conditions should be investigated with each negative control model.
10. Fig 3a and b does not demonstrate the sufficient strategy.
11. What's the role of hydrogel. It should be described clearly.
12. Authors compared treatment effectiveness of each treatment group in Fig. 6. However, the present study design does not demonstrate treatment effectiveness of bacteria (SDV) itself without gel formation.

13. In vitro experiments used human GBM cells, while in vivo therapeutic experiments used mouse GBM. This discrepancy should be addressed.

14. Although authors tried to show the safety of intracranial bacterial application (primarily due to concerns about inducing infection), there is not enough evidence to predict the safety of clinical application. More experiments directly focusing on safety are needed and it should be discussed in the discussion section.

15. Figure 4k shows that GBM cells treated with the constructs release cytokines (IL-2, IL-10, IL-12). It should be clarified that these cytokines likely originate from immune cells.

16. Discussion doesn't present the specific discussion about each important finding. It only has some general information.

Minor comments

1. In Fig. 1(f), Authors described that In vivo distribution of SDVs were assessed via the tail vein injection (not via intracavitary injection). Please clarify the injection route.

2. (line 83) SLNs should be corrected to SLINs.

3. There are a few typographical errors, such as in line 752.

Reviewer #4 (Remarks to the Author): with expertise in GBM, immunotherapy

In their manuscript titled "Stimulation of Tumoricidal Immunity via Bacteriotherapy Inhibits Glioblastoma Relapse", Zhang, Xi and Fu et. al describe their findings using a cavity-injectable bacterium-hydrogel superstructure to target satellites of GBM cells around surgical cavity. They employed an innovative strategy to use bacteriotherapy for immunological cell death, achieving tumor-specificity, lethality and immunogenicity. They encapsulated the IASNDS in a "smart" hydrogel that responds to ATP release following pyroptosis, and further accentuates the immunogenicity of the IASNDS by promoting innate and adaptive immune responses. They show that the gel+IASNDS therapy successfully prevents postoperative relapse of GBM.

The manuscript has immense implications for the field, and can be an innovative strategy to extend survival in the clinic. The experiments to engineer the therapy are robust and the data generated are extensive. The survival effects are impressive. However the interpretation of some of the data needs to be reassessed and the manuscript needs to be reorganized to

avoid repeating concepts and experimental designs. Overall the manuscript should be accepted after addressing some of these minor and major concerns.

1) The authors are encouraged to reorganize the figures or the results section as there is overlap in figure description in figures 1 and 2 (example- SLIN and exosome descriptions).

2) Lines 102-105 mention precise active targeting to tumor cells using CD44 binding. The authors did not present any findings to claim precise targeting. CD44 can be found on many activated lymphocytes as well. Can immune cells also be targeted with the vehicle?

3) Line 159 mentions drug capacity of 4.82% but refers to a schematic of the SLIN. How was 4.82% determined?

4) Similarly line 182 refers to Figure 3b which is a schematic, to say that cleaved caspase 1 converts GSDMD to pore-forming domain N-terminal GSDMD. How was this statement supported? Was increased GSDMD-N confirmed? Could lysis of SDVs alone be sufficient to induce pyroptosis, and would mutated/uncleaved GSDMD in SLINs prevent pyroptosis?

5) Figure 6b is not clear, and the areas pointed by the arrow is not where the tumor/cavity is located. Also, please add to the methods section to describe the resection. Example- what is the 'suction device'? How was it controlled to ensure uniform resection across all groups.

6) The effects of gel + IASNDS in tumor clearance is impressive. Beyond 120 days, what happens to the gel or the IASNDS embedded in it? Can they be visualized in the cavity or in the brain beyond 120 days? If the gel, or the CpG ODN, or the IASNDS is lost/removed/dissolves over time, would tumors recur?

7) Would a tumor rechallenge in the contralateral hemisphere after initial tumor clearance be durable, or would the gel+IASNDS need to be added to the rechallenge site too. Line 374 mentions formation of immune memory but that is not probed in the study. Induction of memory subsets, or rechallenge experiment would support this statement. At the moment, it is hard to delineate memory induction vs effects of prolonged gel+IASNDS exposure.

8) A lot of the conclusions in the study revolved around the incorporation of innate and adaptive immune arms by this therapy. How effective is the non-immune effects of the gel alone or the gel + IASNDS strategy in propagating glioma clearance? (have the authors examined resection + therapy in either immunocompromised mice or mice with immune cell depletions?) This is an important question as patients on chemotherapy might be transiently immunocompromised to fully benefit from this therapy.

Response to reviewers' comments:

We are grateful for the constructive feedback from the reviewers. According to these suggestions, we have provided additional data to validate our conclusion. The manuscript has been substantially revised. Below, please find our point-by-point responses to the reviewers' comments.

Reviewer #1:

This manuscript entitled "Stimulation of Tumoricidal Immunity via Bacteriotherapy Inhibits Glioblastoma Relapse" describes a cavity-injectable bacterium-hydrogel superstructure that targets GBM satellites around the cavity, triggers GBM pyroptosis, and initiates innate and adaptive immune responses, thereby preventing postoperative GBM relapse. This work reports a local bacteriotherapy for stimulating anticancer immunity, particularly for malignancies with a high risk of recurrence. However, considering the innovation and the issues mentioned below, this manuscript does not meet the publication level of Nature Communications.

Major issues:

Q1. It would be better to reorganize the abstract to make it clearer.

A: Thank you for your constructive comments. We have rewritten the abstract to make the information clearer. In addition, we have highlighted the distinctive features and research implications of our study. We also had the manuscript professionally edited by American Journal Experts (AJE) to improve its quality. Additionally, we rearranged the sentence structure and carefully polished the entire manuscript. Both the readability and logical flow were significantly improved to appeal to a broader audience.

Q2. Figure 1f appears to show that the application of SDVs makes brain tissue more porous. The authors should explain the reason behind this observation.

A: Thank you very much for your kind advice. We agree with you that the tumor tissue shown in Figure 1f is more porous. We believe that this difference is mainly caused by 2 factors: i) SDVs induce tumor cell pyroptosis, and pyroptosis increases the number of vacuoles because the typical manifestation of pyroptosis is cellular vacuole formation; ii) tumor immunotherapy may lead to tumor tissue populated sparsely (*ACS nano*, 2018, 12: 12096-12108), mainly due to the elimination of tumor cells, further conducive to immune cell infiltration.

In the revised manuscript (Page 7):

The observed increased porosity in tumor tissue within the SDVs group seems to be associated with the occurrence of pyroptosis in GBM cells, leading to clearance by the immune system.

Q3. In Figure 2i, the authors did not consider the interference caused by components such as nanosized extracellular vesicles secreted by Salmonella, which could affect the size measurement of SLINs. The authors must demonstrate that the size measured is indeed for SLINs and not a mixture containing Salmonella vesicles or other bacterial components.

A: We apologize that we did not present the experimental protocols in the Methods section of the original manuscript. As described on Page 7 of the manuscript for Figure 2i, "SLINs were placed in dialysis bags 10 nm in diameter and incubated with 10^7 bacterial CFU to observe their stability, and their size did not change significantly within one week". During the experiments, to minimize the effects of vesicle production by SDVs, we coincubated SDVs and SLINs by restricting SLINs to dialysis cassettes, which allowed only SDV metabolites and secreted proteins to reach the culture fluid.

In the revised manuscript (Page 21):

We restricted the SLINs to dialysis cassettes of 10 nm in diameter, coincubated them with 1×10^7 CFUs of SDVs, and detected changes in the diameter of the SLINs at different times. The dialysis cassettes selectively allowed SDV metabolites and secreted proteins to enter the culture fluid. At the end of the culture, we used dynamic light scattering (DLS) to determine the particle size of the SLINs and then statistically analyzed the changes.

Q4. It was claimed that ATP-responsive hydrogel may promote long-term immune cell activation. To make the conclusion more solid, there should be a strict non-ATP-responsive hydrogel as a control.

A: As one of the most abundant intracellular metabolites, ATP is an important bio-messenger once it is released into the extracellular environment. In the present study, GBM immunogenic cell death was accompanied by the massive release of ATP, which plays a key role in initiating tumor-specific immune responses by enhancing immunogenicity. Specifically, ATP binds specifically to the ATP receptor P2Y2 located on the surface of antigen-presenting cells (APCs), thereby recruiting APCs to apoptotic sites and promoting the activation of DCs through enhanced accumulation, which in turn recruits and mobilizes T cells to reach the mid-tumor location to inhibit tumor proliferation.

As you suggested, we added a non-ATP-responsive hydrogel as a control group. Compared to those in the non-ATP-responsive hydrogel group, the number of activated tumor-associated APCs in the ATP-responsive hydrogel group was significantly increased.

In the revised manuscript (Page 15):

Furthermore, we investigated the effect of ATP-responsive gel on active immunization, and the non-ATP-responsive IASNDS@gel effect was significantly less vigorous than that in ATP-responsive IASNDS@gel group (Supplementary Fig. 23).

Supplementary Fig. 23:

Supplementary Fig. 23. Flow cytometric analysis of APCs cells within ATP-responsive and non-responsive gels. (b) Shows statistical differences in CD80⁺CD86⁺ cells (n = 6).

Q5. The reviewer suggests that mixing with the hydrogel may limit the movement and diffusion of bacteria. The authors should first test whether the hydrogel itself limits the bacteria from fully exerting their cell invading effects. In animal experiments, they could include a control group where the hydrogel was injected and allowed to form solidification, and then followed by local administration of IASNDS in PBS.

A: We thank the reviewer for the constructive suggestion. We agree with the reviewer that hydrogels may limit bacterial motility compared to PBS. We designed this study initially to address the issue of high local recurrence in GBM patients during clinical management. Postoperative GBM recurs only within 2-3 cm of the operative cavity, and there is no metastasis to distant sites. Localized treatment of GBM using PBS as a carrier for IASNDS does not seem to have a suitable therapeutic effect. The dissemination of IASNDS around brain tissue filled with flowing cerebrospinal fluid (CSF) intracranially may cause potential toxic side effects. After intracavity injection, the precursor solution gelatinized into the hydrogel, from which the cargos were released in a sustained manner, which is more favorable for activating the immune system to eliminate residual GBM cells and prolong or even terminate postoperative recurrence. In conclusion, the therapeutic modality of local hydrogel in the operative cavity is a strategic compromise for coping with local recurrence of GBM after surgery, and the strategy of IASNDS delivery via PBS mentioned by the reviewer can be applied to the local treatment of unresectable GBM; we would like to investigate this further in our future work, and we thank you once again for your enlightening suggestions.

Q6. The in vitro release of CpG ODN was investigated by hydrogel in response to ATP. How is the ATP responsiveness of hydrogel in vivo? This bears significant relevance for achieving effective delivery of IASNDS into the surgical cavity.

A: The intracellular concentration of ATP is in the range of 5-10 millimolar, whereas the concentration of ATP in healthy tissues is typically only measured in nano-molars. The concentration of ATP in the TME of GBM patients with an average of 100 μ M (*Nature Reviews Cancer*, 2018, 18: 601-618; *Cancer letters*, 2003, 198: 211-218). To study the degradation pattern of hydrogel in vivo, we incubated the hydrogel with artificial cerebrospinal fluid containing ATP at concentrations of 0, 0.1 and 100 μ M at 37 °C for 0, 1, 2, 3, 4, 5, 6, and 7 days and detected changes in the weight of the hydrogel.

As shown in Supplementary Fig. 14, the hydrogel exhibited comparable degradation in PBS, and adding ATP obviously accelerated the degradation.

In the revised manuscript (Page 12):

Co-incubation of ATP-responsive hydrogel in artificial cerebrospinal fluid simulating the in vitro environment revealed that the degradation was related to the ATP concentration, with the higher the concentration the faster rate of degradation (Supplementary Fig. 14).

Supplementary Fig. 14:

Supplementary Fig. 14. Changes in gel weight after co-incubation of artificial cerebrospinal fluid with different concentrations of ATP and 30 uL of hydrogel for different periods of time (n = 3).

Q7. Does the UV-curing process employed in hydrogel formation have any impact on the stability and activity of the encapsulated IASNDS?

A: The intensity and duration of UV irradiation are critical factors that determine the inactivation of Salmonella. With a maximum absorption wavelength of 265 nm to kill Salmonella, the irradiation wavelength should reach 10 mW/s/cm² and last for approximately 24 minutes. This condition only kills bacteria on the surface of the gel; for inhibiting the bacteria inside the gel, higher intensities and durations are needed. Hyaluronic acid methacryloyl (HAMA) is a light-curing agent that introduces a methacryl group into the hyaluronic acid chain; HAMA can be used to cure a gel within 10 seconds under UV and visible light. The UV wavelength for our photocrosslinking system was 405 nm, the duration was 10 seconds, and the intensity was 10 mW.s/cm²; therefore, the effect of UV radiation on the IASNDS was negligible.

Q8. It is hard to see bacteria in IASNDS@gel by the SEM image in Figure 5d. High quality images are needed and bacteria inside should be pointed out clearly.

A: We apologize for the lack of clarity in the presentation of IASNDS in Figure 5d that was due to the level of magnification. As you suggested, to better visualize the IASNDS in the hydrogel, we reimaged IASNDS@gel with SEM and labeled the SDVs green.

Revised Fig. 5d:

Fig. 5d: Representative scanning electron microscope image of the hydrogel.

Q9. In Figure 7, the gating strategy should be displayed. For example, the authors should include where the percent of cells of interest are obtained from (i.e. T cells from total viable cells or total CD45⁺ cells).

A: The gating strategy is highly necessary for flow cytometry analysis, and as you suggested, we have added the gating strategy to the original Supplementary Fig. 19 as follows.

Supplementary Fig. 19:

Supplementary Fig. 19. Gating strategy for T-cell analysis. On day 20 post-treatment, groups of mice were humanely euthanized.

Q10. It is confusing that the authors evaluate the change of CD80⁺CD86⁺ T cells in Figure 7f, g. Please clarify.

A: We apologize for the confusion in the description. It should be "CD80⁺CD86⁺ cells", and we have corrected this term in Figure 7f and g. Thank you for your careful review.

Q11. In Figure 7h, the expression profiles display the highest increment in IL-15 expression in IASNDS@gel group. The authors should affirm the protein level of IL-15 expression and discuss the underlying mechanism.

A: Interleukin-15 (IL-15) is a cytokine and signaling molecule involved in the regulation of the immune system. It plays a crucial role in the activation and proliferation of certain immune cells, such as natural killer (NK) cells and T cells. The potential role of IL-15 in cancer therapy has been a hot research topic. IL-15 promotes the central memory T-cell (T_{cm}) phenotype, which is restimulated by antigens, and helps T_{cm} cells clear tumor cells. IL-15 can enhance the antitumor activity of immune

cells and has great potential for cancer immunotherapy. The following are some key points about IL-15 and its potential in cancer treatment:

Stimulation of the immune response: IL-15 can stimulate the activity of NK cells and CD8⁺ T cells, which are important components of the immune system involved in recognizing and destroying cancer cells.

Tumor killing: IL-15 has been found to increase the tumor killing efficiency of immune cells. This could lead to a more effective anticancer immune response.

Combination therapy: Researchers are exploring the use of IL-15 in combination with other immunotherapies, such as checkpoint inhibitors, to improve the overall effectiveness of cancer treatment.

IL-15 prevents activation-induced cell death and induces the generation and steady-state proliferation of longer-lived memory T cells. IL-15 does not increase the number of Treg cells, and it does not act on vascular endothelial cells to cause systemic toxicity. Therefore, IL-15 is currently a promising cytokine for tumor immunotherapy. However, the IL-15 receptor is widely expressed in peripheral tissues, and IL-15 injection causes toxic side effects, limiting its therapeutic efficacy on tumors.

Based on the immunomodulatory activity of IL-15, a highly effective and low-toxicity local therapeutic strategy, IASNDS@gel, was devised in this study. Systemic toxicity can be reduced by preventing the activation and proliferation of peripheral NK cells and T cells while retaining an effective antitumor capacity. The antitumor effect of the IASNDS@gel is dependent on the intrinsic presence of CD8⁺ T cells and the effector molecule IFN- γ . IASNDS@gel treatment significantly increased CD8⁺ T cells within tumor tissue and helped overcome immune checkpoint blockade tolerance. In addition, in a "cold" tumor model, such as GBM, which has little immune cell infiltration, the local treatment strategy of accessing IASNDS@gel can produce synergistic antitumor therapeutic effects, suggesting that IASNDS@gel can help to overcome resistance to targeted therapies to prevent the recurrence of GBM.

References :

1. Guo J, Liang Y, Xue D, et al. Tumor-conditional IL-15 pro-cytokine reactivates anti-tumor immunity with limited toxicity[J]. *Cell Research*, 2021, 31(11): 1190-1198.
2. Chakraborty P, Vaena S G, Thyagarajan K, et al. Pro-survival lipid sphingosine-1-phosphate metabolically programs T cells to limit anti-tumor activity[J]. *Cell reports*, 2019, 28(7): 1879-1893. e7.
3. Kurz E, Hirsch C A, Dalton T, et al. Exercise-induced engagement of the IL-15/IL-15R α axis promotes anti-tumor immunity in pancreatic cancer[J]. *Cancer Cell*, 2022, 40(7): 720-737. e5.
4. Zhou J, Jin L, Wang F, et al. Chimeric antigen receptor T (CAR-T) cells expanded with IL-7/IL-15 mediate superior antitumor effects[J]. *Protein & cell*, 2019, 10(10): 764-769.
5. Kurz E, Hirsch CA, Dalton T, et al. Exercise-induced engagement of the IL-15/IL-15R α axis promotes anti-tumor immunity in pancreatic cancer. *Cancer Cell*. 2022;40(7):720-737.

6. Wrangle JM, Velcheti V, Patel MR, et al. ALT-803, an IL-15 superagonist, in combination with nivolumab in patients with metastatic non-small cell lung cancer: a non-randomised, open-label, phase 1b trial. *Lancet Oncol.* 2018;19(5):694-704.
7. Provasi E, Genovese P, Lombardo A, et al. Editing T cell specificity towards leukemia by zinc finger nucleases and lentiviral gene transfer. *Nat Med.* 2012;18(5):807-815.

Q12. Normal cells and bacteria should be set as a control for the imaging of cellular pyroptosis and bacterial lysis, respectively.

A: Thank you for the constructive suggestion. At the time of the experiment, we added normal cells to the Ctrl group and the SDV and SLIN groups, but none of them exhibited typical cellular pyroptosis. Figure 4b shows the results of the statistical analysis of membrane blebbing data, which revealed a clearer difference than the images in Figure a.

Figure 4a, b:

Figure 4a, b: (a) Visualization of cellular pyroptosis changes induced by the IASNDS under confocal microscopy, showing membrane pores and vacuole formation. (b) Quantification of vacuoles in cells from different treatment groups to assess pyroptosis efficiency (n = 6).

Q13. In addition to visual imaging of bacterial lysis (Fig 3d), alternative methods are recommended to be used for further confirming the occurrence of bacterial lysis.

A: Thank you for the suggestion. After receiving your comments, we tried to find various methods for validation, but the available technologies and methods are limited. For example, we tried to detect DNA fragments from bacteria and use them as a detection marker, but this approach is more challenging in terms of quality control because SDV is an intracellular bacterium. After careful investigation, we found that there is currently no direct method other than direct observation of the bacteria. In addition, as shown in Fig. 3e, g and h, the release of GFP from the bacterial interior into the cytoplasm of GBM cells was evaluated, which implies the bacterial lysis. The data shown in Fig. 4a, b also confirmed that bacterial autolysis caused pyroptosis in GBM cells.

Q14. To further evaluate the immunomodulatory efficacy of the Salmonella-loaded hydrogel system against tumors, the authors are recommended to assess IFN γ ⁺CD8⁺ T cells, M1 macrophages, and M2 macrophages.

A: We thank the reviewer for the constructive suggestion. As suggested by the reviewer, we performed flow cytometry on IFN γ ⁺CD8⁺ T cells, M1 macrophages and M2 macrophages. We found that the ratio of IFN γ ⁺CD8⁺ T cells was significantly elevated, indicating an elevated level of cytotoxic T cells. Similar to the trend of CD80⁺CD86⁺ cell changes, the ratio of M1-type macrophages was significantly elevated in the IASNDS@gel group, whereas the ratio of M2 macrophages was significantly decreased. The above results make the immunomodulatory efficacy of the hydrogel system against tumors more convincing.

In the revised manuscript (Page 14):

Further analysis revealed a significant increase in M1 macrophages (CD80⁺F4/80⁺) and a significant decrease in M2 macrophages (CD206⁺F4/80⁺) in the IASNDS@gel group (Supplementary Fig. 21).

In particular, compared to the Ctrl group, CD8⁺IFN γ ⁺ T cells were significantly higher in the IASNDS@gel treatment group, suggesting a more pronounced anti-tumor immune response within the GBM (Supplementary Fig. 22).

Supplementary Fig. 21:

Supplementary Fig. 21. Flow cytometric results (a) and analysis of statistical differences (b) in intra-tumoral M1 macrophage cells in mice after treatment of in situ hormonal mice with different drugs (n = 6). Flow cytometric results (c) and analysis of statistical differences (d) in intra-tumoral M2 macrophage cells in mice after treatment of in situ hormonal mice with different drugs (n = 6).

Supplementary Fig. 22:

Supplementary Fig. 22. (a) Flow cytometric analysis of CD8⁺IFN γ ⁺ cells in different treatment groups of mice and analysis of statistical results (b, n = 6).

Minor issues:

Q1. The full name is required when the abbreviation first appears, it can help readers to follow and understand. For example, what is the FNR, ICD, CFU and H&E in the paper? Their full names should be added.

A: Thank you for your kind suggestion. We have addressed this issue, and we went through the manuscript in detail to ensure that similar issues do not appear in the revised version.

Q2. The key primers used in this work should be provided in supplementary information. There is nd9 and nd10 in the manuscript, but no corresponding information available in SI. Please check it carefully.

A: The relevant primers and designed gene sequences were double-checked in this manuscript, and the sequences are listed in the file titled "Supplementary Information on SDVs" for easy reference. Thank you for your advice.

Q3. The descriptions should be consistent throughout the entire manuscript. For example, "h" or "hours", " 1×10^8 " or " 1×10^8 ", and many others... Please check thoroughly.

A: We appreciate the reviewer's detailed comments and constructive suggestions. We have thoroughly double-checked the manuscript. In the revised manuscript, the typos and grammar errors have been corrected. Thank you again for your careful review.

Q4. There are a few typos throughout the manuscript. Please doublecheck.

A: Thank you for the comments. We have thoroughly double-checked the manuscript. In the revised manuscript, the typographical and grammar errors have been corrected. Additionally, we rearranged the sentence structure and carefully polished the entire manuscript. Both the readability and logical flow were significantly improved to appeal to a broader audience.

Reviewer #2:

This study introduces a solution for GBM treatment: a bacterium-hydrogel structure administered into the cavity after surgery. This system targets and triggers the death of GBM cells, prompting both innate and adaptive immune responses that deter tumor recurrence. Specifically, this approach utilizes a modified Salmonella delivery vehicle (SDV), derived from weakened Salmonella typhimurium (VNP20009), to seek out and attack GBM cells. Additionally, Salmonella lysis-inducing nanocapsules (SLINs) attached to the SDV stimulate an innate anti-tumor response by releasing bacterial components within the cells. Administered through intracavitary injection using an ATP-responsive hydrogel, this method recruits immune cells, aiding in antigen presentation and initiating an adaptive immune response. Although the therapeutic strategy is remarkable, this study lacks proper control groups in animal studies and assessment of therapeutic efficacy in different GBM models that are highly immunosuppressive. The other suggestions for further improving the study are outlined below:

Q1. The authors conducted immunohistochemistry (IHC) (Figure 1f) and immunofluorescence (IF) (Figure 1g) analyses to illustrate the distribution/accumulation of VNP and the gene-edited VNP variant (referred to as SDV) within the tumor of an orthotopic mouse glioma model subsequent to intravenous injection. However, the IHC (Figure 1f) did not distinctly display the distribution of either VNP or SDV within the tumor. Furthermore, the IF (Figure 1g) failed to definitively establish the accumulation of SDV-GFP within the tumor compared to the normal brain tissue. It is recommended that the authors stain the tumor cells and conduct colocalization studies to ascertain the association between tumor cells and VNP/SDV.

A: We appreciate the reviewer's suggestion. Figure 1f and Figure 1g show the results of IHC and IF staining of brain tissue sections from the same tumor-bearing mouse. Figure 1f shows the location of the tumor to provide positional information about the IF staining. In Figure 1g, we performed nuclear staining (blue, DAPI) on all cells (including GBM cells). The white arrow shows the tumor location, the applied SDV is GFP negative (indicated by the figure note), and the red color (via the GramTM Red⁺ Bacterial Gram Stain Kit) shows the SDV. In the VNP20009 group, since the bacteria were not hypoxic, VNP20009 was scattered and less abundant in the brain tissue. However, the bacteria in the SDV group were enriched mainly in the local hypoxic location of the tumor.

Fortunately, our in vitro experiments confirmed the intracellular localization of SDVs, so we sincerely hope that, with the help of Figure 1h, we can address the reviewer's concerns. In addition, we revised Figure 1g by adding the contour of the tumor tissue, which further confirmed that SDVs can be enriched locally in tumors. We thank you again for your suggestions.

In the revised manuscript (Page 6):

To gain insight into the spatial distribution of SDVs in a syngeneic orthotopic mouse GBM model, we intravenously injected SDVs and VNP20009 (Fig. 1e, f). Compared to VNP20009, the SDVs (red) showed significant aggregation at the tumor site, and dispersion within normal brain tissue was not evident (white arrows, Fig. 1g, bottom).

Revised Figure 1g:

Figure 1g: (g) Additional fluorescence staining of brain tissue sections to observe the SDV distribution (GFP⁻, shown in red) in intracranial GBM tissues (white arrows).

Q2. Notably, the distribution of VNP and SDV in other organs subsequent to intravenous injection was not addressed. The authors should isolate tissue suspensions from these organs to assess the potential dissemination of VNP and SDV in vitro.

A: Thank you for your professional suggestion. We agree that we should observe the distribution of VNP and SDV within the major organs after IV administration. Further, we inoculated heart, liver, spleen, lung and kidney brain suspensions into bacterial culture plates and counted the colonies. We found that the amounts of VNP and SDV in the heart, liver, spleen, lung and kidney brains were very low, and there was no statistically significant difference among them. However, there was a difference in the number of colonies in the suspensions of the brains of the tumor-bearing mice, with a significant increase in the number of bacteria in the SDV group. This may be due to the intravenous injection of bacteria caused the clearance of neutrophils and macrophages, which resulted in a faster clearance of bacteria from the heart, liver, spleen, lungs and kidneys. However, the presence of tumors in the brain disrupted the blood-brain barrier, and the local blood supply to tumors was greater than that to normal brain tissue, allowing more SDV bacteria to evade immune cells.

Supplementary Fig. 28:

Supplementary Fig. 28. Tail vein injection of VNP20009 and SDVs for 2 hours was followed by extraction and bacterial culture of mouse heart, liver, spleen, lung, kidney and brain tissue suspensions (n = 5).

Q3. In Figure 1h, the authors employed 500 μM L-arabinose to induce the lysis of LysE⁺ SDV- GFP; however, this concentration may be excessive. It is suggested to use a significantly lower concentration to evaluate the sensitivity of LysE⁺ SDV-GFP to L-arabinose accurately.

A: We thank the reviewer for the suggestion. As the review's suggestion, we coincubated the L-arabinose with SDV at different concentrations (Supplementary Fig. 3). The lysis efficiency of SDV increased with the L-arabinose concentration and reached a maximum at approximately 500 μM . Consequently, we chose 500 μM as the optimal concentration for inducing SDV lysis in vitro experiments.

Supplementary Fig. 3:

Supplementary Fig. 3. Efficiency curves demonstrating SDV cleavage at various L-arabinose concentrations. Data are presented as the mean \pm S.D.

Q4. The construction details of the plasmids utilized in the study lack specificity. The authors are recommended to provide detailed sequences and relevant information. Additionally, the steps involved in the infection/transduction of the plasmids into the cells/bacteria are not clearly delineated.

A: Thank you for your suggestion. We have put all the sequences of the genes and related primers in the file named "Supplementary Information of SDVs" to facilitate the reviewers' evaluation. Specifically, we have included our transfection procedure in the file.

Q5. The SLIN nanoparticles encapsulating L-arabinose exhibit high stability in vitro. It is essential to elucidate the mechanism triggering the release of L-arabinose from the SLIN nanoparticles in vivo.

A: The pathways and degradation of exosomes in cells are diverse, complex and related to the structure of the exosome and the type of endocytosis. Exosomes first contact the cell membrane and are then endocytosed as endosomes, which may then escape or be degraded. The degradation process mainly includes endosomal and lysosomal degradation (*Science*. 2020, 367: eaau6977; *Nat Rev Neurol*. 2019, 15: 193-203; *Cell Metab*. 2021, 33: 1744-1762). The introduction of synthesized SLINs into cells in this study could occur either by way of SDV-infected cells entering into GBM cells or by way of P-selectin/CD44, which are highly expressed on the surface of SLIN. However, lysosomes are involved in SLIN degradation throughout the process. We applied bafilomycin A1 to inhibit lysosomal acidification and protein degradation and found

that the efficiency of SDV cleavage was significantly inhibited. This finding showed that the drug release from SLIN in this study was lysosome dependent.

In the revised manuscript (Page 9):

Bafilomycin A1 was employed to inhibit lysosomal acidification and protein degradation, the result showed that lysis of SDVs was significantly inhibited, thus suggesting that degradation of SLINs is achieved via the lysosomal pathway (Supplementary Fig. 7).

Supplementary Fig. 7:

Supplementary Fig. 7. Mean fluorescence intensity of QL01#GBM cells after IASNDS-induced autolysis following Bafilomycin A1 inhibition of lysosomal function (n = 9).

Q6. The study indicates that the endosomal sorting complexes required for transport (ESCRT) III-mediated cell membrane repair significantly mitigates tumor cell pyroptosis by repairing and subsequently eliminating gasdermin pores. However, the current system does not impede the repair of the cell membrane mediated by ESCRT III.

A: Thank you for the comment and suggestion. We agree with you that ESCRT III plays a critical role in pyroptosis. We also considered interfering with the formation of ESCRT III to enhance cellular pyroptosis. We analyzed the ESCRT-III complex and found that it only exists transiently and contains both essential and nonessential components. The essential subunits must be assembled in the proper order (Vps20, Snf7, Vps24, then Vps2) for the machinery to function (*FEBS Lett.* 2011, 585: 3191-3196; *Curr. Biol.* 2021, 22: 603-605). The nonessential subunits include Vps60, Did2, and Ist1. Vps20 initiates the assembly of ESCRT-III by acting as a nucleator for Snf7 polymer assembly. Vps24 then binds to Snf7, capping the complex and recruiting Vps2. Vps2 then transports Vps4 into the complex. (*J. Cell Sci.* 2009, 122: 2167-77). The carboxyl termini of most ESCRT-III subunits contain microtubule interaction and transport structural domain motifs (MIMs). These motifs are responsible for binding Vps4 and the AAA-ATPase (*Nat. Cell Biol.* 2011, 13: 394-401; *EMBO J.* 2010, 29: 871-883; *Proc. Natl. Acad. Sci. U. S.A.* 2005, 102: 871-883).

Then, we analyzed the expression of the essential subunits VPS20, SNF7, CHMP3 and VPS2 in GBM cells by database analysis. SNF7 is not expressed in humans, and although VPS20 and VPS2 were slightly highly expressed in patients with GBM, unfortunately, these key genes did not significantly affect the survival of patients. Accordingly, we concluded that interfering with ESCRT-III may have a limited effect

on enhancing GBM pyroptosis. Increasing the protein level of GSDMD is a more direct and effective method than increasing the expression of GSDMD, and we verified by in vivo and in vitro experiments that increasing GSDMD protein level can induce typical cellular pyroptosis (Figure 4a and b).

Supplementary Fig. 27:

Supplementary Fig. 27. (a) We analyzed the expression of ESCRT-III complex-related genes (VPS20, VPS2, CHMP3) in tumor samples from GBM and LGG patients. (b) Difference between VPS20 expression and patient prognosis. (c) Difference between VPS2 expression and patient prognosis.

Q7. Regarding the encapsulation of SDV or IASNDS in CpG-functionalized gel, the individual functionalities of SDV or IASNDS alone remain unclear, particularly in the absence of these agents encapsulated in a normal gel (CpG-unfunctionalized gel).

A: We thank you for your constructive comments, and to demonstrate the therapeutic effect, we added the SDV and IASNDS groups to explore the individual functionalities of SDV or IASNDS. Compared with those in the Ctrl group, the therapeutic effects in the SDV and IASNDS groups did not improve significantly, possibly because the CSF was diluted in the SDV and IASNDS groups. In addition, we investigated the effect of ATP-responsive hydrogels on immune activity, and non-ATP-responsive hydrogels were significantly less effective at preventing immune activation than ATP-responsive hydrogels (Supplementary Figure 23).

In the revised manuscript (Page 13):

Without Gel, SDV or IASNDS alone did not demonstrate significant GBM suppression, primarily because drugs injected locally tend to be rapidly cleared with cerebrospinal fluid circulation.

Revised Figure 6e, f, g:

Figure 6e, f, g: (e) In vivo bioluminescence images were captured (n = 3), and (f) the resulting signal intensity was quantified (n = 10). (g) Mouse survival was assessed through Kaplan-Meier survival curves for each treatment group (n = 10). Data analysis was performed employing the log-rank (Mantel-Cox) test.

Supplementary Fig. 23:

Supplementary Fig. 23. Flow cytometric analysis of APCs cells within ATP-responsive and non-responsive gels. (b) Shows statistical differences in CD80⁺CD86⁺ cells (n = 6).

Q8. IASNDS@gel demonstrates remarkable therapeutic efficacy and extended survival in the GL261 resection model. However, the performance of intracavity injection of IASNDS@gel in unresected GBM models remains undisclosed.

A: Local injection of drugs for intracranial tumors in the absence of surgical treatment is not recommended by clinical guidelines because the skull is a bony structure without adequate space for such a procedure (NCCN Guidelines Version 1.2022). Surgical intervention is still the first choice for GBM management; therefore, we designed a postoperative adjuvant treatment strategy. Glioma with high local recurrence is the most challenging clinical problem, and this study focused on addressing the issue of local recurrence of GBM after surgery. The therapeutic efficacy of intracavity injection of IASNDS@gel in unresected GBM is also a very interesting topic that we would like to explore further in future studies. Thank you for your enlightening suggestions.

Q9. A more comprehensive analysis of the immunoprofiling of other immune cell populations is recommended to provide a more holistic understanding.

A: Thank you for the constructive suggestion. We further performed flow cytometry on IFN γ ⁺CD8⁺ T cells, M1 macrophages and M2 macrophages to enrich the immunoprofile. The ratio of IFN γ ⁺CD8⁺ T cells was significantly elevated in the IASNDS@gel group, indicating that cytotoxic T cells were enriched in GBM tumors. Similarly, the ratio of M1-type macrophages was significantly elevated in the IASNDS@gel group, whereas the ratio of M2 macrophages was decreased correspondingly. The above results strongly indicate that IASNDS@gel-based bacteriotherapy can activate the immune system and significantly enhance antitumor immune cell infiltration.

In the revised manuscript (Page 14):

Further analysis revealed a significant increase in M1 macrophages (CD80⁺F4/80⁺) and a significant decrease in M2 macrophages (CD206⁺F4/80⁺) in the IASNDS@gel group (Supplementary Fig. 21).

In particular, CD8⁺IFN γ ⁺ T cells were significantly higher in the IASNDS@gel treatment group, indicating a more pronounced anti-tumor immune response within the GBM (Supplementary Fig. 22).

Supplementary Fig. 21:

Supplementary Fig. 21. Flow cytometric results (a) and analysis of statistical differences (b) in intra-tumoral M1 macrophage cells in mice after treatment of in situ hormonal mice with different drugs (n = 6). Flow cytometric results (c) and analysis of statistical differences (d) in intra-tumoral M2 macrophage cells in mice after treatment of in situ hormonal mice with different drugs (n = 6).

Supplementary Fig. 22:

Supplementary Fig. 22. (a) Flow cytometric analysis of CD8⁺IFN γ ⁺ cells in different treatment groups of mice and analysis of statistical results (b, n = 6).

Reviewer #3:

This study aims to demonstrate the efficacy of a Salmonella-based system in preventing the recurrence of glioblastoma post-surgery. The authors constructed an attenuated VNP20009 strain of Salmonella with Δ msbB, Δ purI, and Δ xyl deletions to carry two types of plasmids.

The first plasmid contains InvA, controlled by the hypoxic promoter FNR, enabling bacterial invasion into target cells in the hypoxic tumor. The second plasmid contains LysE, controlled by the pBAD promoter, to lyse the bacteria and release their contents. This engineered Salmonella is referred to as the "Salmonella delivery vehicle" (SDV). Additionally, the authors created exosomes, termed "Salmonella lysis-inducing nanocapsules" (SLIN), derived from megakaryocytes. These exosomes carry a fusion protein of exosome-localized BASP1 and the pore-forming protein essential for cellular pyroptosis, gasdermin D BASP1 and the pore-forming protein essential for cellular pyroptosis, gasdermin D (GSDMD). The exosomes are encapsulated with L-arabinose and attached to the Salmonella membrane. Finally, this construct is combined with an immune-activating hydrogel, which couples CpG ODN with a nucleic acid aptamer and cross-links them using UV irradiation to retain the therapeutic factors locally. The authors demonstrate that the system is functional and effective in preventing the invasion and recurrence of brain tumors after surgery.

Although this study was relatively well-designed and the design and concept appeared to be intriguing, this manuscript is not comprehensibly written. Many parts are omitted and dispersed in results, figure legends, and materials & methods. Moreover, specific issues should be discussed in the Discussion section.

Major comments:

Q1. In line 67, "NLRs located on the surface of phagocytes" is incorrect. NLRs is located in the cytosol.

A: We appreciate the reviewer's detailed comment. We have corrected in the revised manuscript, as shown below.

In the revised manuscript (Page 4):

Microorganisms, especially gram-negative bacteria, have the capacity to trigger antitumor immune responses by engaging Toll-like receptors (TLRs) and nucleotide-binding oligomeric receptors (NLRs) of phagocytes.

Q2. This paper is not written in a comprehensible manner. Explanations for many parts are omitted. For instance, there should be an explanation in the results section regarding the purpose of using BASP1 whose full name is not provided at the appropriate place. Additionally, the sentences in lines 112-115 are difficult to understand. The meaning of the 'unlocked' structural configuration is unclear, and the logic claiming that it guarantees safety is also questionable. Moreover, there is mention of adding L-arabinose to exosomes. Meaning behind 'adding it to exosomes' is not clear.

A: We apologize for the unpleasant experience with the manuscript. We appreciate the reviewer's detailed comments and constructive suggestions. After reorganization, both the readability and logical flow of the manuscript were significantly improved. A

professional editor from AJE helped us polish the manuscript. We sincerely hope that the revised manuscript will provide a pleasant reading experience.

In the revised manuscript (Pages 6):

In pursuit of a synergistic therapy employing SDVs, we orchestrated the expression of exosome-localized brain acid soluble protein 1 (BASP1) and gasdermin D (GSDMD), where BASP1 anchors the GSDMD to the surface of the exosome membrane.

The structural configuration not only guarantees the safety of intracranial SDVs application but also augments the capacity of SDVs to induce immunogenic cell death (ICD).

Q3. The paragraph between 118-128 should be rewritten because the explanation of Supplementary figure 1 is not enough and it is not clear how hypoxic and normoxic conditions are made.

A: Thank you for the comment. As suggested by the reviewer, we have revised the description of Supplementary figure 1 and provided a clear description of how anoxic and normoxic conditions are formed. The specific changes are as follows.

In the revised manuscript (Page 6):

In this study, to investigate the capability of SDVs to invade GBM cells, we discovered that auto-lysing SDVs labeled with green fluorescent protein (GFP⁺) proliferated and generated Salmonella-containing vesicles (SCV) in GBM cells (Supplementary Fig. 2). To further validate the response of the SDVs to hypoxia (1% oxygen), the efficacy of the SDVs (GFP⁺) was subsequently investigated in hypoxic and normoxic (21% oxygen) environments, and the results showed that under hypoxic conditions, the survival rate of the SDVs was increased and the fluorescence intensity was enhanced (Fig. 1c, d).

Q4. Figure 1g demonstrated SDV successfully colonize brain tumors, while VNP20009 failed. SDV is a transformed strain of VNP20009 by *invA* and *LysE*. What is the reason of the difference between two strains. *InvA* and *LysE* expression affects the targeting capability?

A: Thank you for the comments. The *invA* gene of SDVs is involved in invasion into host cells and is a key gene involved in enhancing VNP20009 colonization in the tumor microenvironment. To enhance the ability of VNP20009 to recognize hypoxia and maintain responsiveness in the hypoxic tumor microenvironment, we deleted the original *invA* gene of VNP20009 and subsequently fused the hypoxia-inducible promoter FNR with *invA* to achieve responsive regulation of SDVs. A transcriptional activator (FNR) is required for the expression of many genes involved in the anaerobic respiratory pathway, and the fusion of FNR and *invA* can significantly enhance the invasion of SDVs in anaerobic environments. *LysE* is induced to lyse SDVs to increase safety, and pBAD can express target genes in accordance with L-arabinose; herein, our pBAD-*LysE* system can lyse SDVs accordingly, increasing the safety of SDVs.

Q5. SDV and SLINs have gained functions by engineering, but the gain of function was not fully evaluated and verified. For example, *InvA* was employed to better invade the

cancer cells, but the function of InvA was not evaluated. The ratio of invasion of SDV with or without InvA should be quantified.

Additionally, while InvA allows Salmonella to invade epithelial cells, it is important to note that GBM cells, such as GL261, are not epithelial cells. They are glial cells. Does InvA work with glial cells?

In addition, P-selectin was employed in SLIN for binding to CD44 of GBM cells, but binding of P-selectin and CD44 was not verified. Furthermore, the expression level of CD44 in GBM should be presented as compared to normal tissue.

A: Thank you for your suggestion, and we apologize for the lack of clarity regarding the definition of SDV. We defined VNP20009 after modification with InvA and LysE. The expression of InvA was assessed with the hypoxia-inducible promoter FNR constructed with the same plasmid, and the results in Figure 1c-g demonstrated that, compared with VNP20009, SDV could respond to the hypoxic microenvironment of GBM. The cells injected into intracranial in situ tumor-bearing mice were GL261 cells, and Figure 1c shows that SDV can invade GL261 cells.

We reanalyzed the expression of CD44 in GBM cells by using the GEPIA database. CD44 expression was significantly increased in GBM tissue compared with that in normal brain tissue and was approximately 4.6 times greater. Immunohistochemical staining also revealed that the difference in CD44 expression between normal cells and GBM cells was obvious, and GBM cells expressed abundant CD44. Thus, it is feasible to use P-selectin to target CD44 in GBM cells.

In the revised manuscript (Page 5-6):

Exploiting the potential of P-selectin on the surface of megakaryocyte-derived exosomes to bind to CD44, which is highly expressed on the surface of GBM cells, we achieved precise active targeting to tumor cells (Supplementary Fig. 1).

Supplementary Fig. 1:

Supplementary Fig. 1. (a) mRNA expression levels of CD44 in GBM and low-grade glioma (LGG) analyzed by GEPIA database; (b) Immunohistochemical staining showing the expression levels of CD44 in NHA cells; (c) Immunohistochemical staining showing the expression levels of CD44 in QL01# GBM cells. The scale bar is 10 μ m.

Q6. The mechanism of conjugating exosomes (SLIN) to Salmonella VNP20009 (SDV) is not explained.

A: We appreciate the reviewer's insightful suggestion. Click chemistry was employed to couple SLINs to the surface of the SDVs. In this revised manuscript, we have

emphasized the connection between DBCO-SLINs and azide-SDVs. In addition, we have detailed the specific protocols of the experiments in the Methods section as follows:

In the revised manuscript (Page 6):

This fusion protein was strategically positioned on the exosomal membrane, followed by coupling of the exosomes with the SDVs surface via an azide-alkyne cycloaddition reaction. This strategy culminated in the creation of an IASNDS, as depicted in Fig. 1b.

Q7. Figure 1 needs to convey more information. It should explain where GFP is encoded, as well as what is present on the surface and inside the exosomes. In other words, Fig 2f should be demonstrated in Figure 1b.

A: Thank you for your constructive comments, which led us to improve the manuscript. GFP was coexpressed with FNR-InvA; please refer to the Supplementary Material for the specific gene design. We have revised Figure 1b and removed Figure 2f as follows.

In the revised manuscript (Page 6):

Further, the construction of SLINs achieved by encapsulating L-arabinose in GSDMD-EXO through electroporation, SLINs can be degraded intracellularly releasing L-arabinose which could further induce SDVs to express LysE. This fusion protein was strategically positioned on the exosomal membrane, followed by coupling of the exosomes with the SDVs surface via an azide-alkyne cycloaddition reaction. This strategy culminated in the creation of an IASNDS, as depicted in Fig. 1b.

Q8. In this study, gasdermin family proteins, the membrane perforin gasdermin D (GSDMD) was used to trigger cellular pyroptosis, which releases tumor antigens, cytokines, and chemokines that significantly activate antitumor immune responses (line 183). Although Western blot analysis for cellular GSDMD expression was shown in supplementary Fig. 5, therapeutic mechanism using this protein was not explained clearly. Additional experiments or explanations are needed, including comparing treatment effects depending on whether GSDMD is expressed or not.

A: We appreciate the reviewer's comments. GSDMD consists of an N-terminal structural domain (GSDMD-N) and a C-terminal structural domain (GSDMD-C), and GSDMD-N has intrinsic pore-forming activity, which is inhibited by GSDMD-C in the full-length state. After GSDMD is activated by cleavage, the released GSDMD-N binds to membrane lipids and forms pores, leads to the release of cytokines and various cytoplasmic contents (*PNAS*. 2017, 114: 10642-10647; *Nature*. 2016, 535: 111-6; *Immunity*. 2018, 48: 35-44.e6). In this study, the C-terminus of GSDMD was designed to bind BASP1 to facilitate cleavage of the N-terminus. The GSDMD-N could form pores on the surface of GBM cells, which in turn formed vacuoles. The GSDMD-positive and GSDMD-negative IASNDS constructs showed a more significant difference in inducing cellular pyroptosis (Supplementary Fig. 4).

In the revised manuscript (Page 7):

Comparing the difference between SDV equipped with GSDMD (+) SLIN and GSDMD (-) SLIN, respectively, and we found that GSDMD (+) SLIN in combination with SDV could significantly increase cellular pyroptosis (Supplementary Fig. 4).

Supplementary Fig. 4:

Supplementary Fig. 4. Fluorescence staining (a) and statistical analysis (b) of cell membrane vacuole formation in GSDMD high versus low expressing QL01#GBM cells. The scale bar is 10 μ m (n = 5).

Q9. Each step such as the release of L-arabinose from exosomes after Salmonella invades GBM cells, activating the pBAD promoter, and the expression of LysE under these conditions should be investigated with each negative control model.

A: Thank you for your kind and constructive comments. According to the reviewers' suggestions, we examined the expression of the pBAD promoter as well as the expression of LysE in this case. We have shared this information with the reviewers and readers in a file titled "Supplementary Information on SDVs".

Q10. Fig 3a and b does not demonstrate the sufficient strategy.

A: We appreciate the reviewer's detailed suggestion. In this revision, we have added part of the necessary text description to help the readers understand the information. In addition, we have enhanced the contrast between colors to increase the readability of the text. Figure 3a and b depict the changes in the GBM microenvironment induced by IASNDS@gel, and Figure 1a and b show the process and application of IASNDS@gel in vivo and in vitro, respectively, which can help to clarify the entire therapeutic process. Thank you again for your suggestions.

Revised Figure 3a, b:

Editorial note: This figure was created with BioRender.com, released under a Creative Commons Attribution 765 NonCommercial-NoDerivs 4.0 International license

Fig. 3: (a) Illustration depicting the IASNDS system invading GBM cells, triggering cell death, activating antigen-presenting cells, and eliciting an immune response. (b) IASNDS cellular entry, intracellular autocleavage initiation, and tumor cell pyroptosis induction mechanism.

Q11. What's the role of hydrogel. It should be described clearly.

A: Thank you for your kind suggestion. This study aimed to investigate the challenge of high local recurrence following GBM resection. By utilizing hydrogels as carriers for local drug delivery, this approach ensures sustained and controlled release of the drug in the surgical cavity. The brain is surrounded by cerebrospinal fluid (CSF), which decreases the effectiveness of simple local drug injection due to poor fixation in the surgical cavity and susceptibility to CSF dilution. Moreover, hydrogel application prevents extensive drug release, minimizing potential impacts on neurological function. Additionally, hydrogels have immune-activating properties, thereby supporting the potential of combining immunotherapy with GBM treatment.

In the revised manuscript (Page 10):

Since the brain tissue is filled with flowing cerebrospinal fluid which is unfavorable to the local application of the drug, hydrogel as a carrier for local drug delivery can ensure the sustained release in the operative cavity, and can also improve the concentration of the drug locally, avoiding the toxicity to the normal brain tissue.

Q12. Authors compared treatment effectiveness of each treatment group in Fig. 6. However, the present study design does not demonstrate treatment effectiveness of bacteria (SDV) itself without gel formation.

A: Thank you for your constructive comments. To better present the therapeutic effect, we added more information about the SDV and IASNDS groups. Compared with those in the Ctrl group, the therapeutic effects in the SDV and IASNDS groups did not improve significantly, and we speculate that this is mainly because the CSF in the tumor resection cavity was diluted in the SDV and IASNDS groups.

In the revised manuscript (Page 12):

Without Gel, SDV or IASNDS alone did not demonstrate significant GBM suppression, primarily because drugs injected locally tend to be rapidly cleared with cerebrospinal fluid circulation.

Revised Figure 6e, f, g:

Figure 6e, f, g: (e) In vivo bioluminescence images were captured (n = 3), and (f) the resulting signal intensity was quantified (n = 10). (g) Mouse survival was assessed through Kaplan-Meier survival curves for each treatment group (n = 10). Data analysis was performed employing the log-rank (Mantel-Cox) test.

Q13. In vitro experiments used human GBM cells, while in vivo therapeutic experiments used mouse GBM. This discrepancy should be addressed.

A: We thank the reviewer for the comments and constructive suggestions. GL261 cells are a well-characterized mouse HGG cell line, and in vivo application of GL261 cells has been used to investigate the activation of the immune system to inhibit local recurrence after GBM surgery. A well-characterized immune system can better reflect glioma cells in mice in vivo. Humanized immunodeficient mice can only partially mimic human adaptive immunity and cannot take into account both innate and adaptive immunity. This therapeutic paradigm has also been widely used in several published studies (*Science translational medicine*, 2017, 9: eaaf2968; *Cancer Res.* 1970, 30: 2394-2400; *Science Translational Medicine*, 2022, 14: eabn1128; *Nature Nanotechnology*, 2021, 16: 538-548). Therefore, we assume that this is indicative of the actual treatment effect.

Q14. Although authors tried to show the safety of intracranial bacterial application (primarily due to concerns about inducing infection), there is not enough evidence to predict the safety of clinical application. More experiments directly focusing on safety are needed and it should be discussed in the discussion section.

A: We thank the reviewer for the comment. We strongly agree with you that safety was at the top of our list of considerations in designing this project. First, we applied the gel as a carrier locally in the operative cavity, which ensures that the bacteria act locally and reduces the risk of intracranial infection. Second, our autolysis system also ensures safety by inducing the expression of LysE when invading cells, which in turn causes bacterial lysis. We performed blood and cerebrospinal fluid (CSF) bacterial cultures and found that although bacteria were present in the cerebrospinal fluid (CSF), the concentration was extremely low, up to 1 CFU/mL, and did not reach the intracranial infection level, indicating that the strategy of localized application of gel delivery of bacteria is safer.

Supplementary Table 1:

Time (Day)	Ctrl (CFU/mL)		IASNDS@gel (CFU/mL)	
	Blood	CSF	Blood	CSF
1	0	0	0	1
3	0	0	0	0
5	0	0	0	0.5
7	0	0	0	0
9	0	0	0	0

Supplementary Table 1. Bacterial culture of cerebrospinal fluid and blood at different time points after local injection of IASNDS@gel in the operative cavity and counting of colonies (n = 5).

Q15. Figure 4k shows that GBM cells treated with the constructs release cytokines (IL-2, IL-10, IL-12). It should be clarified that these cytokines likely originate from immune cells.

A: We apologize for the misleading description. In this study, our intention was to show that IASNDS treatment induced the secretion of immune activation-associated cytokines and chemokines by coincubation with THP-1 macrophages. We modified the description of Figure 4k as follows.

In the revised manuscript (Page 10):

Further studies showed that IASNDS treatment caused a significant increase in the levels of cytokines and chemokines associated with immune activation of THP-1 cells after co-incubation with QL01#GBM cells (Fig. 4k).

Q16. Discussion doesn't present the specific discussion about each important finding. It only has some general information.

A: Thank you for the comment. Following the reviewer's suggestion, we have substantially revised the Discussion section to include specific discussions related to each of our key findings. In the revised manuscript, we discuss concerns about safety and the prospects for clinical translation. We hope these improvements greatly enhance the specificity and clarity of the Discussion section, and improve the overall quality of the manuscript.

Minor comments

Q1. In Fig. 1(f), Authors described that In vivo distribution of SDVs were assessed via the tail vein injection (not via intracavitary injection). Please clarify the injection route.

A: We apologize for the confusion in the description. We have added a description of the tail vein injection procedure to the "Materials and Methods" section; please refer to the changes below.

In the revised manuscript (Page 21):

Mice were immobilized before tail vein injection. The tail vein was rinsed with ethanol to induce congestion and dilation in preparation for injection. The injection started from the end of the tail vein, choosing an entry point close to the thumbnail and placing the needle at an angle of approximately 30° to the vessel. The needle is gently inserted into the skin with the beveled side facing up. After insertion, align the needle parallel to the vessel, keeping the hand steady to prevent fluid leakage.

Q2. (line 83) SLNs should be corrected to SLINs.

A: We apologize for the mistakes caused by carelessness, which we have corrected in the revised manuscript; additionally, we have conducted a careful review to minimize the occurrence of such issues.

Q3. There are a few typographical errors, such as in line 752.

A: Thank you for pointing out the typos. We carefully reviewed the revised manuscript and a professional editor carefully proofread our manuscript. We hope that our efforts will minimize the occurrence of such problems.

Reviewer #4:

In their manuscript titled “Stimulation of Tumoricidal Immunity via Bacteriotherapy Inhibits Glioblastoma Relapse”, Zhang, Xi and Fu et. al describe their findings using a cavity-injectable bacterium-hydrogel superstructure to target satellites of GBM cells around surgical cavity. They employed an innovative strategy to use bacteriotherapy for immunological cell death, achieving tumor-specificity, lethality and immunogenicity. They encapsulated the IASNDS in a “smart” hydrogel that responds to ATP release following pyroptosis, and further accentuates the immunogenicity of the IASNDS by promoting innate and adaptive immune responses. They show that the gel+IASNDS therapy successfully prevents postoperative relapse of GBM.

The manuscript has immense implications for the field, and can be an innovative strategy to extend survival in the clinic. The experiments to engineer the therapy is robust and the data generated are extensive. The survival effects are impressive. However the interpretation of some of the data needs to be reassessed and the manuscript needs to be reorganized to avoid repeating concepts and experimental designs. Overall the manuscript should be accepted after addressing some of these minor and major concerns.

Q1. The authors are encouraged to reorganize the figures or the results section as there is overlap in figure description in figures 1 and 2 (example- SLIN and exosome descriptions).

A: Thank you for pointing out this issue. In the revised manuscript, we removed Figure 2f and integrated the relevant content into Figure 1b to avoid duplication of content. We also conducted several rounds of review and discussion to ensure that each figure represents our research in a reasonable manner. Thank you again for your careful review.

In the revised manuscript (Page 8):

This circuit was achieved by encapsulating L-arabinose in GSDMD-EXO via electroporation, forming nano-capsules named SLINs with 4.82% drug loading capacity measured by HPLC. Further characterization of these SLINs revealed that their morphology did not change significantly after loading with L-arabinose, and the particle size was approximately 100 nm (Fig. 2f, g).

Revised Figure 1b:

Editorial note: This figure was created with BioRender.com, released under a Creative Commons Attribution 765 NonCommercial-NoDerivs 4.0 International license

Fig. 1: (b) Illustration detailing the preparation of the IASNDS and synergistic localized treatment with the hydrogel for surgical cavity application.

Q2. Lines 102-105 mention precise active targeting to tumor cells using CD44 binding. The authors did not present any findings to claim precise targeting. CD44 can be found on many activated lymphocytes as well. Can immune cells also be targeted with the vehicle?

A: We appreciate the reviewer's detailed comment. The overexpression of CD44 in gliomas seems to be widely confirmed, and related studies have been published (*Cancer Cell*. 2010, 18: 655-668; *Cell Stem Cell*. 2014, 14: 357- 369; *Proc Natl Acad Sci U S A*. 2020, 117: 11432-11443). In addition, many studies have utilized CD44, which is highly expressed on the surface of gliomas, as a target for precision therapy (*Sci Adv*. 2023, 9: eade5321; *J Control Release*. 2023, 354: 572-587; *Biomaterials*. 2019, 198: 63-77). Analysis of the database revealed that CD44 expression was significantly increased in GBM cells tissue compared with normal brain tissue. We also performed immunohistochemical staining and found that the difference in CD44 expression between normal cells and GBM cells was also very obvious, and the surface of GBM cells greatly overexpressed CD44. Thus, targeting CD44 on the surface of GBM cells via P-selectin is feasible. The strategy used in this study was localized postoperative treatment where the drug was localized to the operative cavity, which also greatly reduced the side effects on normal brain tissue.

In the revised manuscript (Page 5):

Exploiting the potential of P-selectin on the surface of megakaryocyte-derived exosomes to bind to CD44, which is highly expressed on the surface of GBM cells, we achieved precise active targeting to tumor cells (Supplementary Fig. 1).

Supplementary Fig. 1:

Supplementary Fig. 1. (a) mRNA expression levels of CD44 in GBM and low-grade glioma (LGG) analyzed by GEPIA database; (b) Immunohistochemical staining showing the expression levels of CD44 in NHA cells; (c) Immunohistochemical staining showing the expression levels of CD44 in QL01# GBM cells. The scale bar is 10 μ m.

Q3. Line 159 mentions drug capacity of 4.82% but refers to a schematic of the SLIN. How was 4.82% determined?

A: We apologize for the misleading description in our original submission. HPLC detection of L-arabinose was performed in this study. A C18 reversed-phase column was used with a UV-Vis detector set at a maximum wavelength (λ_{max}) of 260 nm.

In the revised manuscript (Page 8):

This circuit was achieved by encapsulating L-arabinose in GSDMD-EXO via electroporation, forming nanocapsules named SLINs with 4.82% drug loading capacity measured by HPLC.

Q4. Similarly line 182 refers to Figure 3b which is a schematic, to say that cleaved caspase 1 converts GSDMD to pore-forming domain N-terminal GSDMD. How was this statement supported? Was increased GSDMD-N confirmed? Could lysis of SDVs alone be sufficient to induce pyroptosis, and would mutated/uncleaved GSDMD in SLINs prevent pyroptosis?

A: Thank you for the comment. GSDMD consists of an N-terminal structural domain (GSDMD-N) and a C-terminal structural domain (GSDMD-C), and GSDMD-N has intrinsic pore-forming activity, which is inhibited by GSDMD-C in the full-length state. After GSDMD is activated by cleavage, the released GSDMD-N binds to membrane lipids and forms pores, which leads to the release of cytokines and various cytoplasmic contents (*PNAS*. 2017, 114: 10642-10647; *Nature*. 2016, 535: 111-6; *Immunity*. 2018, 48: 35-44.e6). In this study, the C-terminus of GSDMD was designed to bind BASP1 to facilitate cleavage of the N-terminus. GSDMD-N could form pores on the surface of GBM cells, which in turn formed vacuoles. The GSDMD-positive and GSDMD-negative IASNDS constructs showed a more significant difference in inducing cellular pyroptosis (Supplementary Fig. 4).

Bacteria can induce tumor cells to undergo pyroptosis, but the expression of GSDMD in tumor cells is not sufficient. The survival time of patients with high GSDMD expression was significantly greater than that of patients with low GSDMD expression; therefore, we considered improving the therapeutic efficacy of GSDMD by delivering GSDMD to GBM cells. As shown in the survival curves in Figure 6g, high expression of GSDMD in SLINs significantly prolonged the survival time of the mice.

Supplementary Fig. 26:

Supplementary Fig. 26. (a) GEPIA database analysis of GSDMD mRNA differences between GBM and LGG. (b) Difference in survival time analyzed between patients with high and ground expression of GSDMD.

Q5. Figure 6b is not clear, and the areas pointed by the arrow is not where the tumor/cavity is located. Also, please add to the methods section to describe the resection. Example- what is the 'suction device'? How was it controlled to ensure uniform resection across all groups.

A: We apologize for the problem with the layout of the image. The issue described was rectified in the revised manuscript, and a red circle was added to indicate the location of the tumor. The method of surgical resection has been added to the "Methods" section; please refer to the description below. The suction device was inspired by the suction used in the clinical surgical resection of tumors. This suction was modified from a phlebotomy suction device, and the tip of the suction tip was modified to create a 30 μ L cavity, which ensured that the volume of the injected drugs was consistent.

In the revised manuscript (Page 12):

To establish the GBM excision mouse model, a portion of the intracranial GBM in mice was removed with forceps under a microscope on day 10, and then the tumor was suctioned by using a suction device to form a cavity that could accommodate 30 μ L volume, hydrogel with different formulations were subsequently injected into the cavity (Supplementary Fig. 16 and 17).

Revised Figure 6b:

Fig. 6: (b) Upon reaching day 10 after initial inoculation, tumor-bearing GL261 mice underwent surgical tumor resection, followed by hydrogel injection. Subsequently, brain tissues from the excised mice were subjected to euthanasia.

Supplementary Fig. 16:

Supplementary Fig. 16. Suction device for mouse GBM resection, simulating clinical surgical suctioning with specified parameters.

Q6. The effects of gel + IASNDS in tumor clearance is impressive. Beyond 120 days, what happens to the gel or the IASNDS embedded in it? Can they be visualized in the cavity or I happens to the gel or the IASNDS embedded in it? Can they be visualized in the cavity or in the brain beyond 120 days? If the gel, or the CpG ODN, or the IASNDS is lost/removed/dissolves over time, would tumors recur?

A: We appreciate the reviewer's comments. This is very thought-provoking and interesting. The release of the drug locally in the operative cavity from the IASNDS@gel was sustained for approximately 1 week. GBM recurs after surgery only at the local surgical margins, so local treatment can activate the immune system to clear tumor cells for a period after surgery. Our cavity-injectable hydrogel system exhibited excellent degradability in vivo and sustained drug release properties.

The key to inhibiting local recurrence after surgery is to activate the immune system to clear the tumor, and immune memory is also critical for prolonged tumor immunity (*Nature*. 2019, 565, 240-245; *Cancer Discov*. 2021, 11, 2248-2265; *Proc Natl Acad Sci USA*. 2022, 119, e2111003119.) The results of flow cytometry analysis of immune memory cells showed that the application of IASNDS@gel could increase the percentage of immune memory cells, which shows that IASNDS@gel treatment has therapeutic effects on short-term clearance of tumor cells and sustained tumor immunity.

In the revised manuscript (Page 15):

On further analysis, the activation and increase in immune memory cells may play a very important role in this process (Supplementary Fig. 24).

Supplementary Fig. 24:

Supplementary Fig. 24. Differences of immune memory cells in tumor tissues of mice in different treatment groups (n = 6).

Q7. Would a tumor rechallenge in the contralateral hemisphere after initial tumor clearance be durable, or would the gel+IASNDS need to be added to the rechallenge site too. Line 374 mentions formation of immune memory but that is not probed in the study. Induction of memory subsets, or rechallenge experiment would support this statement. At the moment, it is hard to delineate memory induction vs effects of prolonged gel+IASNDS exposure.

A: We thank the reviewer for the comment. We agree with the reviewer. Activation of the immune system by IASNDS@gel eliminated residual surgical GBM cells, which prolongs the survival time of mice for more than 120 days. A rechallenge experimental study in hormonal mice revealed that IASNDS@gel-treated mice were virtually incapable of regrowing tumors, and the survival of these mice was significantly prolonged (Supplementary Fig. 25).

In the revised manuscript (Page 15):

We further re-challenged mice originally bearing GBM in the right side, and found that when the mice were treated with IASNDS@gel for 20 days and then transplanted with GBM cells to the left side, the IASNDS@gel treatment significantly inhibited the growth of GBM, and the survival time of the mice was also prolonged (Supplementary Fig. 25).

Editorial note: Panel a of this figure was created with BioRender.com, released under a Creative Commons Attribution 765 NonCommercial-NoDerivs 4.0 International license

Supplementary Fig. 25:

Supplementary Fig. 25. Rechallenge the experiment. (a) Diagram of the experimental design model. (b) H&E staining of mouse brain tissue sections. (c) Kaplan-Meier survival analysis of rechallenged mice.

Q8. A lot of the conclusions in the study revolved around the incorporation of innate and adaptive immune arms by this therapy. How effective is the non-immune effects of the gel alone or the gel + IASNDS strategy in propagating glioma clearance? (have the authors examined resection + therapy in either immunocompromised mice or mice with immune cell depletions?) This is an important question as patients on chemotherapy might be transiently immunocompromised to fully benefit from this therapy.

A: We appreciate the reviewer's detailed comments and constructive suggestions. . The recommended treatment option for patients in whom GBM has been detected for the first time is still surgical resection, with the option of postoperative radiotherapy and chemotherapy. The designed bacteriotherapy is an adjuvant to surgical treatment and may be a possible alternative to radiotherapy and chemotherapy in the future, mainly because recurrence occurs in almost 90% of GBM patients after surgery, and the recurrence is limited to only 2-3 cm of the surgical cavity. In this study, we performed surgical resection in situ without radiotherapy in mice that had normal immune systems. Patients diagnosed with GBM are evaluated preoperatively in the clinic, and surgical intervention is not recommended if they have reduced immune cells or impaired immune function, therefore, we consider that immunodeficient mice may not be a good reflection of clinical treatment (NCCN Guidelines Version 1.2022).

REVIEWER COMMENTS

Reviewer #1 (Remarks to the Author):

The authors have addressed my concerns as requested. The paper can be considered for acceptance and publication after the authors checking the legends and adding statistical information in SI.

Reviewer #3 (Remarks to the Author):

1. This paper is not informative. It lacks essential information in the figure legends, such as time, cell lines, quantity, and other necessary research methodology details, which undermines the reliability of the data. Authors should provide all critical information comprehensively and understandably when writing scientific papers. However, this paper fails to do so. Furthermore, some questions of mine were not answered in the letter and revised manuscript. For instance, the un-answered question was that “Additionally, while InvA allows Salmonella to invade epithelial cells, it is important to note that GBM cells, such as GL261, are not epithelial cells. They are glial cells. Does InvA work with glial cells?”
2. For example, in Fig. 1c-h, the authors did not specify which cells were used for the in vitro and in vivo experiments.
3. SDV was not engineered to enhance tumor-targeting capacity. Instead, it was designed to invade host cells in hypoxic conditions and to undergo self-lysis in the presence of L-arabinose. It is crucial to clarify why VNP20009 cannot survive in the hypoxic microenvironment of GBM. Furthermore, the study does not present evidence regarding the extent of hypoxia in the orthotopic GBM model used. The mechanisms by which SDV accumulates and survives in GBM tissue should also be clearly described. Additionally, the reasons for VNP20009's failure to accumulate and survive in GBM were not clearly understood.
4. The authors mention in the legend of Fig. 1h that ‘initiation of self-cleavage (line 766)’. What does this statement imply? Is it indicative of SDV undergoing lysis? Furthermore, the

finding of Fig 1h reveals that GFP is localized within the SCV in the absence of LysE, whereas it is distributed throughout the cytoplasm in the presence of LysE? Is it right? If this is accurate, what is the mechanism through which GFP is released from the SCV following bacterial lysis. The authors are encouraged to elucidate the observations related to the absence of LysE as compared to its presence more comprehensively in the results section.

5. The explanation of how IASNDS operates is lacking across pages 7 and 8. Specifically, the function of GSDMD is still not described in the revised version. Furthermore, the mechanism by which IASNDS invades GBM cells, followed by the degradation of SLIN, the release of arabinose, and the interaction of GSDMD with caspase, remains unexplained. How does GSDMD within the SVC interact with caspase? The authors must convincingly explain the mechanism of action after IASNDS invades the cell.

Reviewer #4 (Remarks to the Author):

In their revised manuscript, Zhang et al. further support their findings and conclusions from their original manuscript by incorporating new data, addressing concerns with interpretation of the previous data, and rearranging the manuscript. The manuscript is a strong candidate for acceptance to Nature Communications.

By using a cavity-injectable bacterium-hydrogel superstructure, they show GBM satellites can be targeted around surgical cavity. By incorporating bacteriotherapy for immunological cell death, they achieve tumor-specific lethality, while inducing an immune response in both the adaptive and innate compartments. Their gel+IASNDS therapy was successful in preventing postoperative relapse of GBM, and they show in their revised manuscript that the strong memory CD8 response induced by the therapy is enough to control new tumor growth on the contralateral side of the brain (rechallenge model), without needing further gel+IASNDS injections.

The work has great implications for the treatment of GBM as it incorporates an innovative strategy that can replace radiation and chemotherapy. The experiments conducted are

thorough and the interpretations in the current submission are robust. Overall, the manuscript should be accepted for publication.

Response to reviewer's comments:

We are grateful for the constructive feedback from the reviewers. According to these suggestions, the manuscript has been appropriately revised. Below, please find our point-by-point responses to the reviewer's comments.

Reviewer #1 (Remarks to the Author):

The authors have addressed my concerns as requested. The paper can be considered for acceptance and publication after the authors checking the legends and adding statistical information in SI.

A: We appreciate the reviewer's effort in reviewing this manuscript.

Reviewer #3 (Remarks to the Author):

Q1. This paper is not informative. It lacks essential information in the figure legends, such as time, cell lines, quantity, and other necessary research methodology details, which undermines the reliability of the data. Authors should provide all critical information comprehensively and understandably when writing scientific papers. However, this paper fails to do so. Furthermore, some questions of mine were not answered in the letter and revised manuscript. For instance, the un-answered question was that "Additionally, while InvA allows Salmonella to invade epithelial cells, it is important to note that GBM cells, such as GL261, are not epithelial cells. They are glial cells. Does InvA work with glial cells?"

A: Thank you for your valuable feedback. We sincerely appreciate your time and effort

in providing constructive criticism. We have carefully considered your comments and have made the necessary revisions to address the issues you raised. Regarding the concerns about the lack of essential information in the figure legends, we apologize for any oversight on our part. We have revised the figure legends to include important details such as time, cell lines, quantity, and other necessary research information to enhance the clarity and reliability of the data presented (Fig. 1 Fig. 3 and Fig. 4), please refer to the revised manuscript with red color in the “Manuscript_Markup Version”.

The perspective that GBM cells are glial cells seems to be debatable. Recent studies have found that GBM originates from neural stem cells and neural precursor cells, and there is also significant tumor heterogeneity in GBM, which shows that GBM cells possess their own characteristics, and it seems to be inappropriate to categorize them as normal endothelial cells or glial cells (Neuro-oncology, 2022, 24, 1494-1508; Cell stem cell, 2020, 26, 48-63; Neuro-oncology, 2010, 12, 422-433). In our experimental studies we demonstrated that hypoxia-driven expression of InvA in SDV enhances targeting of GBM cells, which is mainly due to the hypoxic tumor microenvironment of GBM (Fig. 1e, f and g). Once again, we sincerely appreciate your feedback, and we hope that the revisions we have made adequately address your concerns.

Q2. For example, in Fig. 1c-h, the authors did not specify which cells were used for the in vitro and in vivo experiments.

A: We appreciate the reviewer’s suggestion. As outlined in the "Materials and Methods" section, we utilized luciferase-labeled GL261 cells specifically for the in vivo

experiments involving intracranial transplantation. In Fig. 1c and h, we indeed employed GL261 cells, as detailed in the revised figure legend.

Q3. SDV was not engineered to enhance tumor-targeting capacity. Instead, it was designed to invade host cells in hypoxic conditions and to undergo self-lysis in the presence of L-arabinose. It is crucial to clarify why VNP20009 cannot survive in the hypoxic microenvironment of GBM. Furthermore, the study does not present evidence regarding the extent of hypoxia in the orthotopic GBM model used. The mechanisms by which SDV accumulates and survives in GBM tissue should also be clearly described. Additionally, the reasons for VNP20009's failure to accumulate and survive in GBM were not clearly understood.

A: In comparison to VNP20009, SDV with *fnr-InvA* exhibits heightened responsiveness to the microenvironment, enabling effective targeting of the GBM microenvironment. The characteristic hypoxic conditions within GBM are well-established, as discussed on page 6 of the manuscript, lines 123-127. While acknowledging the importance of hypoxia in GBM, our current focus centers on investigating the efficacy of SDV-based surgical cavity treatment for GBM. Following your inspiration, we recognize the significance of exploring the correlation between hypoxia extent and SDV infiltration in future studies.

The mechanism underlying accumulation and survival of SDVs in GBM primarily stems from the introduction of *fnr-InvA*. Additionally, the local immunosuppressive microenvironment contributes to inhibiting immunological clearance of SDVs, a

hypothesis supported by relevant experiments (Fig. 1c-h). In addition, we assume that the inability of VNP20009 to accumulate and thrive in GBM is due to its inability to colonize the hypoxic microenvironment after injection, and subsequently be cleared by immune cells upon exposure to normoxic tissue.

In the revised manuscript (Page 7):

The mechanism of SDV accumulation and survival in GBM is due to the introduction of *fnr-InvA*, and the local immunosuppressive microenvironment also inhibited the immunological clearance of SDVs.

Q4. The authors mention in the legend of Fig. 1h that ‘initiation of self-cleavage (line 766)’. What does this statement imply? Is it indicative of SDV undergoing lysis? Furthermore, the finding of Fig 1h reveals that GFP is localized within the SCV in the absence of LysE, whereas it is distributed throughout the cytoplasm in the presence of LysE? Is it right? If this is accurate, what is the mechanism through which GFP is released from the SCV following bacterial lysis. The authors are encouraged to elucidate the observations related to the absence of LysE as compared to its presence more comprehensively in the results section.

A: Thank you for your thorough assessment. The statement signifies that the presence of 500 μ M L-arabinose can trigger autolysis of SDV. We hypothesize that upon bacterial lysis, the dynamic biogenesis of SCV is disrupted, leading to the cessation of two Type III secretion systems (T3SS1 & T3SS2). Consequently, GFP fluorescence, previously confined within the SCV, diffuses into the GBM cytoplasm. This phenomenon is vividly

observed as green fluorescence diffuses into the GBM cytoplasm, while in the control group, green fluorescence remains localized within the bacteria. In the revised manuscript, we have expanded upon the observations concerning the absence and presence of LysE more comprehensively, providing further clarity on these findings.

In the revised manuscript (Page 28):

Visualization of the green fluorescence distribution of the SDVs (GFP+) under pBAD-LysE (-) and pBAD-LysE (+) conditions induced by 500 μ M L-arabinose for 24 hours in the cell culture medium. The intracellular GFP distribution in the pBAD-LysE (+) condition indicates initiation of SDV autolysis in GL261 cells.

Q5. The explanation of how IASNDS operates is lacking across pages 7 and 8. Specifically, the function of GSDMD is still not described in the revised version. Furthermore, the mechanism by which IASNDS invades GBM cells, followed by the degradation of SLIN, the release of arabinose, and the interaction of GSDMD with caspase, remains unexplained. How does GSDMD within the SVC interact with caspase? The authors must convincingly explain the mechanism of action after IASNDS invades the cell.

A: Thanks to your comments, we have provided a detailed description of the mechanism of IASNDS and further illustrate the mechanism of action of each component in manuscript (Fig. 3a, b; Page 8). The targeting mechanism of IASNDS is mainly achieved through the high expression of CD44 on the surface of tumor cells (Supplementary Fig. 1).

The manuscript (Page 7):

Free GSDMD does not diffuse into GBM cells mainly due to its large molecular weight and negative surface charge, and GSDMD is unable to trigger cellular pyroptosis owing to concealment of its pore-forming domains³⁸. To address this challenge, *Salmonella typhimurium* was used for intracellular delivery of GSDMD. Due to the properties of the intracellular bacterium in the SDV in the IASNDS, the SLINs on its surface are brought inside the GBM cell, and after inducing pyroptosis, they further activate innate and adaptive immunity to combat postoperative relapse (Fig. 3a).

The IASNDS arrives in the cell and forms *Salmonella*-containing vesicles (SCVs), which can rapidly multiply and prevent cellular clearance (Supplementary Fig. 5). SLINs can be degraded by GBM cells and release GSDMD and L-arabinose, slowly initiating the SDV lysis process, which can further activate caspase 1 to cleaved caspase 1. Then, cleaved caspase 1 converts GSDMD into pore-forming domain N-terminal GSDMD to bind to the GBM membrane and subsequently trigger cellular pyroptosis (Fig. 3b). As a type of ICD, pyroptosis releases tumor antigens, cytokines, and chemokines that significantly activate antitumor immune responses.

In the revised manuscript (Page 7):

Here, GSDMD was attached to the exosome surface by the exosome membrane protein BASP1 to increase the content of GSDMD in tumor cells (Fig. 2a).

Reviewer #4 (Remarks to the Author):

In their revised manuscript, Zhang et al. further support their findings and conclusions

from their original manuscript by incorporating new data, addressing concerns with interpretation of the previous data, and rearranging the manuscript. The manuscript is a strong candidate for acceptance to Nature Communications.

By using a cavity-injectable bacterium-hydrogel superstructure, they show GBM satellites can be targeted around surgical cavity. By incorporating bacteriotherapy for immunological cell death, they achieve tumor-specific lethality, while inducing an immune response in both the adaptive and innate compartments. Their gel+IASNDS therapy was successful in preventing postoperative relapse of GBM, and they show in their revised manuscript that the strong memory CD8 response induced by the therapy is enough to control new tumor growth on the contralateral side of the brain (rechallenge model), without needing further gel+IASNDS injections.

The work has great implications for the treatment of GBM as it incorporates an innovative strategy that can replace radiation and chemotherapy. The experiments conducted are thorough and the interpretations in the current submission are robust.

Overall, the manuscript should be accepted for publication.

A: We appreciate the reviewer's comments and encouragement.

REVIEWERS' COMMENTS

Reviewer #3 (Remarks to the Author):

The authors have addressed my concerns as requested. The paper can be considered for acceptance and publication